

# Bulk entanglement entropy in perturbative excited states

**Alexandre Belin**[1][⋆], **Nabil Iqbal**[2][†] **and Sagar F. Lokhande**[1][∘]

**1** Institute for Theoretical Physics, University of Amsterdam,
Science Park 904, 1098 XH Amsterdam, The Netherlands
**2** Centre for Particle Theory, Department of Mathematical Sciences,
Durham University, South Road, Durham DH1 3LE, UK

⋆ a.m.f.belin@uva.nl
† nabil.iqbal@durham.ac.uk
∘ sagar.f.lokhande@gmail.com

## Abstract

We compute the bulk entanglement entropy across the Ryu-Takayanagi surface for a one-particle state in a scalar field theory in $AdS_3$. We work directly within the bulk Hilbert space and include the spatial spread of the scalar wavefunction. We give closed form expressions in the limit of small interval sizes and compare the result to a CFT computation of entanglement entropy in an excited primary state at large $c$. Including the contribution from the backreacted minimal area, we find agreement between the CFT result and the FLM and JLMS formulas for quantum corrections to holographic entanglement entropy. This provides a non-trivial check in a state where the answer is not dictated by symmetry. Along the way, we provide closed-form expressions for the scalar field Bogoliubov coefficients that relate the global and Rindler slicings of $AdS_3$.

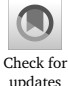

# 1   Introduction

Large $N$ and strong/weak dualities typically map quantum features of a given theory to classical structures in its dual. A particularly dramatic example of this phenomenon is given by holography and the AdS/CFT correspondence. At infinite $N$, holography maps quantum *entanglement* in a field theory – perhaps the most intrinsically quantum-mechanical property that one might imagine – to something as simple as the classical area of a minimal surface $\gamma_A$ in its gravitational dual [1]:

$$S_{\text{EE}} = \frac{A(\gamma_A)}{4G_N}. \tag{1}$$

The simplicity and universality of this relationship, which relates two very primitive ideas on each side of the duality, has led to many practical applications as well as considerable insight into the inner workings of holography.

If we now move away from strictly infinite $N$, the situation is slightly more complicated. The gravitational theory is no longer strictly classical, and the first quantum correction to equation (1) in the bulk has been argued by Faulkner, Lewkowycz and Maldacena (FLM) to be [2]:

$$S_{\text{EE}} = \frac{A(\gamma_A)}{4G_N} + S_{\text{bulk}}(\Sigma_A), \tag{2}$$

where $\Sigma_A$ is a bulk spatial region contained between $\gamma_A$ and the boundary, and $S_{\text{bulk}}(\Sigma_A)$ is the *bulk* entanglement entropy of low energy perturbative degrees of freedom in the bulk, viewed as an effective quantum field theory. In other words, at finite $N$ boundary entanglement splits apart into a classical area term and a *bulk* entanglement term. The generalization of this formula to all orders in $N^{-1}$ was discussed in [3,4].

Jafferis, Lewkowycz, Maldacena and Suh (JLMS) extended equation (2) to an operator equation in [5] which reads

$$K_{\text{CFT}} = \frac{\hat{A}}{4G_N} + K_{\text{bulk}}, \tag{3}$$

where $K$ is the modular Hamiltonian and $\hat{A}$ should now be understood as an area *operator*. This formula is valid as an operator equation on a subset of the full Hilbert space usually called the code subspace. The code subspace is the set of all states that can be accessed in the bulk perturbative quantum field theory around a fixed classical background.

The appearance of a bulk entanglement term (or a bulk modular Hamiltonian) means that the holographic dictionary is now sensitive to the bulk perturbative Hilbert space, and equation (2) is thus a key tool in understanding how quantum degrees of freedom in the bulk can be encoded in the boundary [6, 7]. It also uncovered a deep connection between quantum gravity and quantum error correction [8, 9]. It is intriguing to speculate that the possibility of splitting the CFT entanglement entropy into a classical geometric piece and a bulk matter entanglement piece is a key feature for the emergence of (what we conventionally call) geometry from microscopic field-theoretical degrees of freedom.

However, despite the conceptual importance of equation (2), to our knowledge it has been used in relatively few explicit computations of quantum corrections to entanglement entropy (see however [10, 11]). Many direct calculations of quantum corrections to entanglement entropy instead study states that can be prepared by a path integral and proceed using a partition function approach [12] (see also [13–17] for CFT calculations) rather than directly tracing out states in the bulk Hilbert space. We believe this is largely due to the difficulty of explicitly computing bulk entanglement entropies.

Thus motivated, in this work we develop the tools required to directly apply the FLM relation in a non-trivial setting. In particular, we study a scalar field $\phi$ in the bulk of $AdS_3$, with a mass $m$ that is $\mathcal{O}(1)$ in units of the AdS radius. We consider the lowest-energy excited state with a single $\phi$ quantum sitting in the center of global $AdS_3$. This state has an extended wave-function that fills the bulk of AdS; the energy of this wavefunction backreacts on the bulk geometry to create a conical deficit-like geometry, where the singularity at the center is smoothened out by the Compton wavelength of the particle. On the boundary, this state is simply the primary operator $O$ dual to $\phi$.

We then compute both sides of equation (2) for a (small) interval of length $\theta$ in this excited state, subtracting the contribution from the vacuum. We thus obtain the leading quantum correction which is $\mathcal{O}(1)$ in the large $c$ limit. On the boundary this is done using $CFT_2$ techniques and large $c$-factorization. In the bulk, we compute both the shift in the minimal area due to the backreaction of the particle as well as the bulk entanglement entropy. The latter turns out to be a non-trivial exercise in bulk quantum field theory, as the spatial spread of the $\phi$ wavefunction polarizes the entanglement already present in the vacuum, creating extra entanglement across the bulk of $AdS_3$.

We compute this shift in entanglement entropy by explicitly constructing the perturbative reduced density matrix in the excited state, decomposing the action of the single-particle creation operator $a^\dagger$ across the entangling region $\Sigma_A$ and its complement. This requires the construction of the appropriate Bogoliubov coefficients, which we present in closed form. To the best of our knowledge, this is the first computation of these coefficients. The computation of the entanglement entropy is technically difficult, and the particular techniques that we use are feasible due to the symmetries of the conformal vacuum, which is tied to the vacuum modular Hamiltonian being a local operator in the bulk. In principle our techniques allow an explicit computation for all $\theta$, but in practice we were unable to perform the required analytic continuations away from the small $\theta$ limit, and so we express our final answers as an expansion in interval size.

Both from the bulk and boundary, we find the leading difference in entanglement entropy between the vacuum and the excited state to be:

$$\Delta S_{\text{EE}} = h\frac{\theta^2}{3} - \left(\frac{\theta}{2}\right)^{8h} \frac{\Gamma\left(\frac{3}{2}\right)\Gamma(4h+1)}{\Gamma\left(4h+\frac{3}{2}\right)} + \cdots, \tag{4}$$

where $\theta$ is the length of the interval and $h$ the holomorphic conformal dimension of $O$. We have kept both the leading analytic and non-analytic dependence on $\theta$. In the bulk and the boundary, the two types of terms above have a different physical origin, as we explain. This provides a non-trivial test of the FLM relation in a state where nothing is protected by symmetry. Interestingly, this agreement requires a non-trivial cancellation between the two terms in equation (2), which is a manifestation of a bulk first law of entanglement.

While this agreement is satisfying, we anticipate that the machinery that we develop in performing this computation will have further concrete applications in an understanding of how information is encoded in the bulk and the boundary. We note that from the point of view of quantum field theory in the bulk, we are studying how the presence of a *localized* and perturbative quantum excitation shifts the entanglement pattern of the vacuum; we are not aware of much previous work in this regard (see however [18] where similar ideas were explored in the context of a local quench). We would also like to emphasize that we are focussing on quantum low-energy excitations of the bulk, which is different from the heavy state computation [19] or the coherent states [20, 21].

This paper is organized as follows. In Section 2 we begin by presenting the CFT computation of the entanglement entropy using large-$c$ factorization. In Section 3 we explain in detail the bulk setup, constructing the single-particle state and computing its backreaction on the geometry. We also explain the different coordinate systems that will be used and compute the Bogoliubov coefficients used to express the scalar field in each of them. In Section 4 we study the vacuum modular Hamiltonian in the single particle state; this is an instructive application of some of the technology and it provides intermediate results that are necessary for the final calculation. Finally in Section 5 we explicitly compute the reduced density matrix of the single-particle state and compute the entanglement entropy. We conclude with some directions for future work. Details of the computation of the Bogoliubov coefficients are relegated to an Appendix.

## 2 CFT Calculation

### 2.1 Entanglement Preliminaries

Consider a quantum system in a state $|\psi\rangle$ with Hilbert space $\mathcal{H}$. Now imagine dividing the Hilbert space into two subsystems, $A$ and its complement $\bar{A}$. To characterise the entanglement between $A$ and its complement, we define the reduced density matrix

$$\rho_A \equiv \mathrm{Tr}_{\bar{A}} |\psi\rangle \langle\psi| . \tag{5}$$

There are several measures of entanglement and perhaps the most famous of them all is the entanglement entropy, which is the Von Neumann entropy of the reduced density matrix

$$S_{\mathrm{EE}} = -\mathrm{Tr}\rho_A \log \rho_A . \tag{6}$$

There are other measures of entanglement, such as for example the Rényi entropies

$$S_n \equiv \frac{1}{1-n} \log \mathrm{Tr}\rho_A^n . \tag{7}$$

The use of the Rényi entropies is twofold. First, the set of all Rényi entropies is the most complete basis-independent description of the reduced density matrix $\rho_A$ since the Rényi entropies are the moments of its eigenvalue distribution. Second, a direct computation of the entanglement entropy is usually difficult in quantum field theory but one can circumvent this problem

by the replica trick [22, 23]. One computes the Rényi entropies for all $n$ and then analytically continues in $n$ to obtain the entanglement entropy:

$$S_{\text{EE}} = \lim_{n \to 1} S_n \,. \tag{8}$$

In this paper, we will be interested in calculating entanglement entropies in two-dimensional conformal field theories, which we now describe in more detail.

## 2.2 Entanglement in CFT$_2$

We start by describing the general machinery for computing the entanglement entropy of excited states in arbitrary two-dimensional CFTs, and only later specialize to holographic theories. We will review the important points of [23, 24]. Consider a two dimensional CFT $\mathscr{C}$ in a state $|\psi\rangle$. We take the CFT to live on a circle of unit radius parameterized by a coordinate $\varphi$, and we define subsytem $A$ to be the spatial interval

$$A : \varphi \in [0, \theta] \,. \tag{9}$$

We would like to understand the entanglement between the region $A$ and its complement for a class of states $|\psi\rangle$. The states we will consider are those obtained by acting with a primary operator on the vacuum, namely

$$|\psi\rangle = O(0)|0\rangle \,, \tag{10}$$

for a Virasoro primary operator $O$ with dimensions $(h, \bar{h})$. The dual state is given by[1]

$$\langle\psi| = \lim_{z \to \infty} \langle 0| O(z) z^{2h} \bar{z}^{2\bar{h}} \,. \tag{11}$$

A direct computation of entanglement entropy is often too difficult so one generally proceeds with the replica trick. For two-dimensional CFTs, this is done by considering the orbifold CFTs $\mathscr{C}^{\otimes n}/\mathbb{Z}_n$, in which case the Rényi entropies are defined as correlation functions of local operators (twist operators) [22, 23, 25]:

$$S_n = \frac{1}{1-n} \log \langle O^{\otimes n}| \sigma_n(0,0) \bar{\sigma}_n(0,\theta) |O^{\otimes n}\rangle \,. \tag{12}$$

It is important to note that the operator $O$ that created the state is now raised to the $n$-th power [24]. This occurred because the replica trick instructs us to prepare $n$ copies of the state. In quantum field theory, the entanglement and Rényi entropies are UV-divergent quantities. We will be interested in computing UV-finite quantities, and for this reason we will compute the difference of entanglement entropies between the excited state and the vacuum. We have

$$\Delta S_n \equiv S_n^{\text{ex}} - S_n^{\text{vac}} = \frac{1}{1-n} \log \frac{\text{Tr}\rho_A^n}{\text{Tr}\rho_{A,\text{vac}}^n} = \frac{1}{1-n} \log \frac{\langle O^{\otimes n}| \sigma_n(0,0) \bar{\sigma}_n(0,\theta) |O^{\otimes n}\rangle}{\langle 0| \sigma_n(0,0) \bar{\sigma}_n(0,\theta) |0\rangle}. \tag{13}$$

It is possible to analyse this correlation function directly in the orbifold theory using known properties of the twist operators [25] but we will proceed in a slightly different manner. We will perform a uniformization map that takes us to the covering space. In the process, we get rid of the twist operators and will only have a correlation function in the original CFT $\mathscr{C}$. Consider the transformation from $w$ on the cylinders to $z$ on the plane

$$z = \left( \frac{e^w - 1}{e^w - e^{i\theta}} \right)^{\frac{1}{n}} \,. \tag{14}$$

---

[1]In this paper, we will consider uncharged and therefore Hermitian operators but it would be straightforward to generalize to charged operators.

This first maps the correlation function to the plane through the exponential map, and then uniformizes the twist operators. The $2n$ operators $O$ are now inserted on the complex plane at the positions

$$z_k = e^{-i(\theta - 2\pi k)/n}, \qquad \tilde{z}_k = e^{2\pi i k/n}, \qquad k = 0, ..., n-1. \tag{15}$$

The difference of Rényi entropies therefore becomes

$$\Delta S_n = \frac{1}{1-n} \log \left[ e^{-i\theta(h-\bar{h})} \left( \frac{2}{n} \sin\left[ \frac{\theta}{2} \right] \right)^{2n(h+\bar{h})} \langle \prod_{k=0}^{n-1} O(\tilde{z}_k) O(z_k) \rangle \right]. \tag{16}$$

The twist operators have disappeared, and we are left with a $2n$-point correlation function in $\mathscr{C}$. Furthermore, the conformal factors due to the twist operators (which are responsible for the UV-divergence) are cancelled between the numerator and denominator, yielding a UV-finite answer.

In general CFTs, the computation of this correlation function is difficult as it involves knowledge of all the CFT data of the theory. We will now specialize to large $c$ CFTs with holographic duals where things will simplify.

## 2.3 Holographic CFTs

As we have seen, the Rényi entropies of certain excited states are given by $2n$-point correlation functions which in general are hard to compute. Fortunately, we are interested in holographic large $c$ CFTs in which case the problem drastically simplifies. The reason for this is large $c$ factorization [26]. In large $c$ CFTs, one distinguishes between two classes of operators: first, there are single-trace operators, whose dual is a propagating field in the bulk. Then, there are multi-trace operators which are not dual to new bulk fields, but rather to multiparticle states in the bulk.

Large $c$ factorization in such theories is a statement about OPE coefficients. It states that

$$C_{O_1 O_2 O_3} \sim \frac{1}{\sqrt{c}}, \tag{17}$$

for $O_{1,2,3}$ three single trace operators. On the other hand, multi-trace operators may have an OPE coefficient that is order one. In the large $c$ limit, the different single-trace operators don't talk to each other and are just generalized free fields. In this paper, we will consider single particle states in the bulk and we will therefore take the primary operator $O$ to be single-trace. Due to large $c$ factorization, the leading order contribution to the $2n$-point function (16) is simply given by all the Wick contractions

$$\langle \prod_{k=0}^{n-1} O(\tilde{z}_k) O(z_k) \rangle = \sum_{g \in S_{2n}} \prod_{j=1}^{n} \langle O(z_{g(2j-1)}) O(z_{g(2j)}) \rangle + \mathcal{O}\left( \frac{1}{c} \right). \tag{18}$$

Note that the leading contribution is $\mathcal{O}(1)$, as expected since it is the first correction and the leading term of the entanglement entropy scales like $c$. We will not discuss the subleading terms in equation (18), but they are very interesting since they probe beyond the generalized free field regime. Some progress in understanding them from the bulk was made in [27]. From now on, we will restrict to the order one correction so $\Delta S_n$ should be thought of as $\Delta S_n|_{\mathcal{O}(1)}$. Using the expression for the $z_k$, we have

$$\Delta S_n = \frac{1}{1-n} \log \left[ e^{-i\theta(h-\bar{h})} \left( \frac{2}{n} \sin\left[ \frac{\theta}{2} \right] \right)^{2n(h+\bar{h})} \frac{1}{2^n n!} \sum_{g \in S_{2n}} \prod_{j=1}^{n} \frac{1}{z_{g(2j-1),g(2j)}^{2h} \bar{z}_{g(2j-1),g(2j)}^{2\bar{h}}} \right], \tag{19}$$

where we have defined

$$
\frac{1}{z_{j,k}^{2h} \bar{z}_{j,k}^{2\bar{h}}} \equiv
\begin{cases}
\frac{e^{(2i\varphi/n - 2\pi i(j+k)/n)(h-\bar{h})}}{(2|\sin\frac{\pi(j-k)}{n}|)^{2(h+\bar{h})}}, & j,k \leq n \\
\frac{e^{(i\varphi/n - 2\pi i(j+k)/n)(h-\bar{h})}}{\left(2|\sin\left(\frac{\pi(j-k)}{n} - \frac{\varphi}{2n}\right)|\right)^{2(h+\bar{h})}}, & j \leq n, \quad k > n \\
\frac{e^{(i\varphi/n - 2\pi i(j+k)/n)(h-\bar{h})}}{\left(2|\sin\left(\frac{\pi(j-k)}{n} + \frac{\varphi}{2n}\right)|\right)^{2(h+\bar{h})}}, & k \leq n, \quad j > n \\
\frac{e^{(-2\pi i(j+k)/n)(h-\bar{h})}}{(2\sin|\frac{\pi(j-k)}{n}|)^{2(h+\bar{h})}}, & j,k > n.
\end{cases}
\tag{20}
$$

We can now perform the product over $j$ in equation (19). Regardless of the permutation $g$, the phase factors will cancel out exactly and we obtain

$$
\Delta S_n = \frac{1}{1-n} \log\left[\left(\frac{1}{n}\sin\left[\frac{\theta}{2}\right]\right)^{2n(h+\bar{h})} \mathrm{Hf}(M_{ij})\right],
\tag{21}
$$

where $\mathrm{Hf}(M)$ is the Haffnian of a matrix $M$ defined by

$$
\mathrm{Hf}(M) = \frac{1}{2^n n!} \sum_{g \in S_{2n}} \prod_{j=1}^{n} M_{g(2j-1), g(2j)},
\tag{22}
$$

and

$$
M_{ij} =
\begin{cases}
\frac{1}{(|\sin\frac{\pi(i-j)}{n}|)^{2(h+\bar{h})}}, & i,j \leq n \\
\frac{1}{\left(|\sin\left(\frac{\pi(i-j)}{n} - \frac{\varphi}{2n}\right)|\right)^{2(h+\bar{h})}}, & i \leq n, \quad j > n \\
\frac{1}{\left(|\sin\left(\frac{\pi(i-j)}{n} + \frac{\varphi}{2n}\right)|\right)^{2(h+\bar{h})}}, & j \leq n, \quad i > n \\
\frac{1}{(|\sin\frac{\pi(i-j)}{n}|)^{2(h+\bar{h})}}, & i,j > n.
\end{cases}
\tag{23}
$$

This is the exact expression for the Rényi entropy of primary single-trace states to order $c^0$. Unfortunately, it is hard to perform the analytic continuation to obtain the entanglement entropy for general $h$ and $\bar{h}$. We salute the courage of the authors of [28–31], who managed to analytically continue the answer for $h + \bar{h} = 1$. The answer reads

$$
\Delta S_{\mathrm{EE}} = -2\left(\log\left(2\sin\frac{\theta}{2}\right) + \Psi\left(\frac{1}{2\sin\frac{\theta}{2}}\right) + \sin\frac{\theta}{2}\right) \qquad h + \bar{h} = 1,
\tag{24}
$$

where $\Psi$ is the digamma function. We were not able to extend these results to general conformal dimension since the relation between the Hafnian and the determinant used in [28] only appears to work for $h + \bar{h} = 1$. For this reason, we will now consider the small interval limit so that we can perform the analytic continuation and obtain expressions for the entanglement entropy.

## 2.4 The Small Interval Limit

The small interval limit $\theta \ll 1$ corresponds to the OPE limit in the $2n$-point correlation function, as the $n$ pairs of operator see the two coming close together. In this limit, the calculation of the entanglement entropy was first obtained in [32] and we review the main steps of the computation in what follows[2]. For the remainder of the paper, we will also set $h = \bar{h}$ and

---

[2]Note that the authors computed the relative entropy between the vacuum and the ground state but it is straightforward to obtain the change in the entanglement entropy from there by adding in the change in the modular hamiltonian (26)

consider only scalar excitations. The leading term is given by the identity contribution to all OPE contractions in equation (16)[3]. It reads

$$\Delta S_n \approx \frac{1}{1-n} \log \left[ \left( \frac{\sin\left[\frac{\theta}{2}\right]}{n \sin\left[\frac{\theta}{2n}\right]} \right)^{4nh} \right].$$
(25)

The analytic continuation of this can easily be computed and gives

$$\Delta S_{\text{EE}} \approx 2h \left( 2 - \theta \cot\left(\frac{\theta}{2}\right) \right).$$
(26)

The right-hand side of this expression is fixed by conformal invariance and is therefore universal for all CFTs. In fact, it is exactly equal to the expectation value of the vacuum modular hamiltonian in the excited state we considered [32]. We wish to go beyond this order and probe the generalized free fields as we discussed in the previous subsection. From the OPE point of view, all the other Wick contractions we were summing over in equation (19) come from the contribution of multi-trace operators. These are the only contributions of order $\mathcal{O}(c^0)$ in the large $c$ expansion. For an example of how they can account for the other Wick contractions in four-point functions, see [33].

The lightest operator that can appear in the $O \times O$ OPE is the operator $: O^2 :$ with conformal dimension $4h$. The OPE coefficient is given by [17]

$$C_{OOO^2} = \sqrt{2}.$$
(27)

The exchange of $O^2$ by any two pairs of the $2n$-point function gives a contribution

$$2 \left( \frac{\sin\left[\frac{\theta}{2}\right]}{n \sin\left[\frac{\theta}{2n}\right]} \right)^{4nh} \left( \left( \sin\frac{\theta}{2n} \right)^{8h} \sum_{k=1}^{n-1} \frac{n-k}{(\sin\frac{\pi k}{n})^{8h}} \right).$$
(28)

Using the periodicity of the sine function, this can be rewritten as

$$\left( \frac{\sin\left[\frac{\theta}{2}\right]}{n \sin\left[\frac{\theta}{2n}\right]} \right)^{4nh} \left( \left( \sin\frac{\theta}{2n} \right)^{8h} n \sum_{k=1}^{n-1} \frac{1}{(\sin\frac{\pi k}{n})^{8h}} \right).$$
(29)

The analytic continuation of this expression was obtained (again with a lot of courage) in [34]. It reads

$$n \sum_{k=1}^{n-1} \frac{1}{(\sin\frac{\pi k}{n})^{8h}} = (n-1) \frac{\Gamma\left(\frac{3}{2}\right) \Gamma(4h+1)}{\Gamma\left(4h+\frac{3}{2}\right)} + \mathcal{O}((n-1)^2).$$
(30)

We can then obtain the expression for the entanglement entropy

$$\Delta S_{EE} = 2h \left( 2 - \theta \cot\left(\frac{\theta}{2}\right) \right) - \left( \sin\frac{\theta}{2} \right)^{8h} \frac{\Gamma\left(\frac{3}{2}\right) \Gamma(4h+1)}{\Gamma\left(4h+\frac{3}{2}\right)} + \cdots.$$
(31)

This is the result we will reproduce in the bulk. Note that there are higher-order terms in $\theta$ that we have neglected, arising from the exchange of higher dimension operators; thus technically speaking the above expression should only be viewed as determining the leading analytic and non-analytic terms in a series expansion.

---

[3]Note that the there might be leading contributions, depending on the lowest dimension operator that $\mathcal{O}$ couples to, which gives a contribution $\theta^{2\Delta_{\text{exch}}}$. In particular, for the tricritical Ising model a term coming from the exchange of another operator than the identity would dominate (if the excited operator is $\sigma$, it can exchange an $\epsilon$ operator with $\Delta = 1/5$).

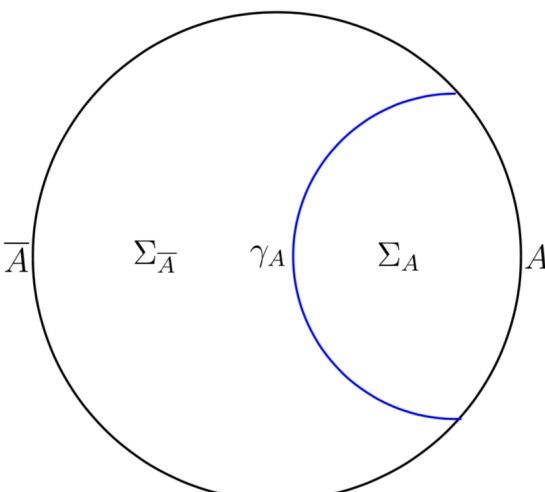

Figure 1: The $t = 0$ slice of the bulk and boundary. The boundary is separated into two regions $A$ and $\bar{A}$. The bulk is also separated into two regions $\Sigma_A$ and $\Sigma_{\bar{A}}$, split by the geodesic $\gamma_A$ that hangs from the boundary entangling surface.

## 3 Bulk Kinematics

We now turn to the bulk. The previous section described the calculation of the left hand-side of the FLM formula (2) for the entanglement entropy of an interval $A$ of angle $\theta$ in an excited state. In the following three sections, we will describe the right-hand side of that formula, i.e. the bulk computation. Recall that in the classical gravity limit this is simply given by the length of a geodesic $\gamma_A$ that hangs down into the bulk dividing space into two regions, the entangling region $\Sigma_A$ and its complement. This is represented in Fig. 1.

By the normal rules of AdS/CFT the scalar operator $\mathcal{O}$ of the previous section is dual to a scalar field $\phi$ propagating in AdS$_3$, and in particular the CFT state corresponding to the primary (10) is dual to a single-particle state in global AdS$_3$.
The combined action of the scalar field and gravitational sector is

$$S = \int d^3x \sqrt{-g} \left( \frac{1}{16\pi G_N} (R+2) - \frac{1}{2}(\nabla\phi)^2 - \frac{1}{2}m^2\phi^2 \right). \tag{32}$$

Here we have set the AdS$_3$ radius to 1. The boundary scalar has equal holomorphic and anti-holomorphic scaling dimension $h = \bar{h}$, and the relation between the bulk mass and boundary dimension is the usual

$$m^2 = 4h(h-1). \tag{33}$$

The equation of motion of the scalar field is

$$(\nabla^2 - m^2)\phi(x) = 0. \tag{34}$$

### 3.1 Global Coordinates

We begin by discussing the mode expansion of the bulk scalar field and the properties of the single-particle state. In this section we work in global AdS$_3$, whose metric is

$$ds^2 = -(r^2+1)dt^2 + \frac{dr^2}{r^2+1} + r^2 d\varphi^2. \tag{35}$$

where $\varphi \sim \varphi + 2\pi$. We will denote the bulk state with this metric and with the scalar field (and all boundary gravitons) in their ground state by $|0\rangle$; this is dual to the vacuum of the CFT on $S^1 \times \mathbb{R}$.

### 3.1.1 Mode Expansion of the Scalar Field

We turn now to the scalar field, which we canonically quantize in modes on this background as

$$\phi(t, r, \varphi) = \sum_{m,n} \left( a_{m,n} e^{-i\Omega_{m,n}t} f_{m,n}(r, \varphi) + a_{m,n}^\dagger e^{i\Omega_{m,n}t} f_{m,n}^*(r, \varphi) \right). \tag{36}$$

We expand the spatial wavefunctions $f_{m,n}(r, \varphi)$ in Fourier modes labeled by $m \in \mathbb{Z}$ along the $\varphi$ circle as

$$f_{m,n}(r, \varphi) = e^{2\pi i m \varphi} f_{m,n}(r), \tag{37}$$

where the remaining quantum number $n$ labels the modes in the radial direction. The mode functions $f_{m,n}(r)$ are solutions to the bulk wave equation that are smooth at the origin of AdS$_3$ at $r = 0$ and are normalizable as $r \to \infty$. These two conditions result in a discrete spectrum for the frequency $\Omega$ [35]

$$\Omega_{m,n} = 2h + |m| + n, \qquad n \in \mathbb{N}. \tag{38}$$

The ladder operators obey the usual relations

$$[a_{m,n}, a_{m',n'}^\dagger] = \delta_{n,n'}\delta_{m,m'}. \tag{39}$$

As usual, this is equivalent to the canonical commutation relations between $\phi(x)$ and its conjugate momentum provided that the spatial wavefunctions are orthogonal with the following normalization[4]:

$$2\Omega_{m,n} \int dr\, d\varphi \sqrt{-g}\, g^{tt}(r) f_{n,m}(r, \varphi) f_{n',m'}^*(r, \varphi) = \delta_{n,n'}\delta_{m,m'}. \tag{40}$$

### 3.1.2 Excited State and Backreaction

We will focus on the bulk state that is dual to the boundary CFT primary; this is the lowest-energy scalar excitation in the bulk, and so has $m = n = 0$. The wavefunction for this mode can be easily found by demanding that it be annihilated by the bulk isometries corresponding to $L_1, \bar{L}_1$ [36], and is simply:

$$f_{0,0}(r) = \frac{1}{\sqrt{2\pi}} \frac{1}{(1 + r^2)^h}. \tag{41}$$

States with nonzero $m, n$ are global conformal descendants of this primary state. The bulk excited state $|\psi\rangle$ is now defined to be:

$$|\psi\rangle = a_{0,0}^\dagger |0\rangle. \tag{42}$$

This corresponds to a single $\phi$ particle sitting in the center of global AdS$_3$ with spatial wavefunction given by equation (41).

We note however that the definition of the bulk state as equation (42) is only complete at strictly $G_N = 0$. At finite $G_N$ the backreaction of this particle will distort the geometry in the bulk; and so more properly the creation operator in equation (42) should also include this correction to the gravitational degrees of freedom.

---

[4]The left-hand side of this expression is the usual Klein-Gordon inner product.

At first order in $G_N$ this is captured by the classical shift in the metric sourced by the expectation value of the stress tensor in the 1-particle state on pure AdS$_3$. In other words we will solve the classical Einstein equation

$$R_{\mu\nu} - \frac{1}{2}g_{\mu\nu}R - g_{\mu\nu} = 8\pi G_N \langle\psi|T_{\mu\nu}|\psi\rangle, \tag{43}$$

to first order in $G_N$, treating the right-hand side as a small perturbation. The stress tensor $T_{\mu\nu}$ for the scalar field is given by the usual expression,

$$T_{\mu\nu} =: \partial_\mu\phi\,\partial_\nu\phi - \frac{1}{2}g_{\mu\nu}\big((\nabla\phi)^2 + m^2\phi^2\big): . \tag{44}$$

We have defined the stress tensor to be normal ordered with respect to the ladder operators in equation (39), and thus in the AdS$_3$ vacuum we have

$$\langle 0|T_{\mu\nu}(x)|0\rangle = 0. \tag{45}$$

We briefly digress to discuss the significance of this choice. In general, the scalar field will present a UV-divergent zero-point energy, which will renormalize the bulk cosmological constant; the choice (45) means that we have added a counter-term to the bulk action to precisely cancel this infinite shift. Furthermore loops of the scalar will also renormalize the bulk Newton's constant (see e.g. [37] for explicit computations in a similar context). Note however that none of these considerations will affect our calculation, as all such UV-sensitive effects are state-independent and will cancel when we consider the difference in entanglement entropy between $|\psi\rangle$ and the vacuum.

The stress tensor is quadratic in $\phi$; thus generically it depends on a double sum over modes. However we will evaluate it only in a particle-number eigenstate; in this case it depends only on the terms that have a product of a creation and an annihilation operator. It is then straightforward to calculate the expectation value of the stress tensor in the state (42)

$$
\begin{aligned}
\langle\psi|T_{tt}(r)|\psi\rangle &= \frac{2h(2h-1)}{\pi}\frac{1}{(1+r^2)^{2h-1}}, \\
\langle\psi|T_{rr}(r)|\psi\rangle &= \frac{2h}{\pi}\frac{1}{(1+r^2)^{2h+1}}, \\
\langle\psi|T_{\varphi\varphi}|\psi\rangle &= \frac{2hr^2}{\pi}\frac{(1-2h)r^2+1}{(1+r^2)^{2h+1}}.
\end{aligned}
\tag{46}
$$

The one particle state thus creates a lump of energy at the center of AdS of width given by the weight $h$. This is shown in Fig. 2

We now solve the Einstein equations (43) to first order in $G_N$. We pick the following gauge for the metric:

$$ds^2 = -(r^2 + G_1(r)^2)dt^2 + \frac{dr^2}{r^2 + G_2(r)^2} + r^2 d\varphi^2, \tag{47}$$

From here it is straightforward to calculate the first order corrections to the metric to be

$$G_1(r) = 1 - 8G_N h + O((G_N h)^2), \tag{48}$$

$$G_2(r) = 1 - 8G_N h\left(1 - \frac{1}{(r^2+1)^{2h-1}}\right) + O((G_N h)^2). \tag{49}$$

Note that the spatial sections of this metric are smoothened out versions of the well-studied conical deficit geometries. At infinity the deficit angle of the cone is $8\pi G_N h$: as usual in 3d gravity, the asymptotic deficit angle measures the energy of the particle inside [38]. However at the origin the metric is completely regular, as the quantum spread of the wavefunction (41) smears the extra bulk energy over a finite radius. We will see that our calculation of the quantum corrections to the entanglement entropy will be sensitive to the structure of this regularization.

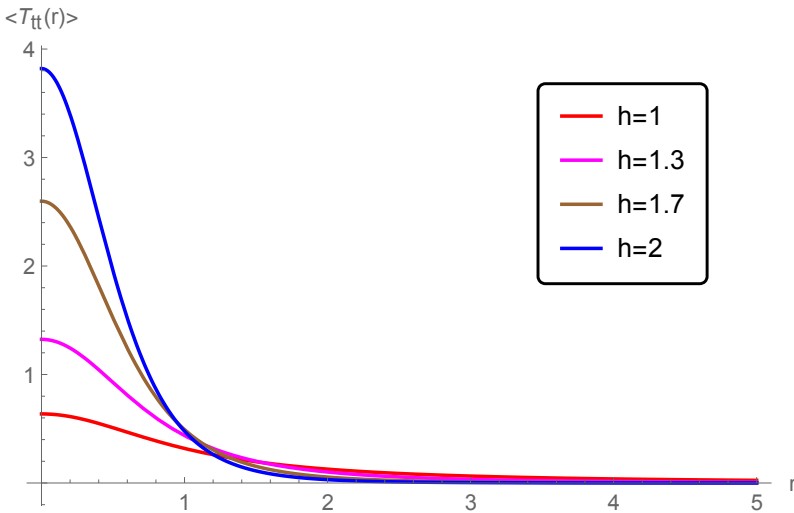

Figure 2: The expectation value of $T_{tt}$ in the one-particle state for different conformal weights.

## 3.2 AdS-Rindler Coordinates

We will now describe a different coordinate system that will be better suited to compute the entanglement entropy of the bulk scalar field. We divide a spatial slice through global AdS$_3$ into two regions, the entangling region $\Sigma_A$ and its complement. We will now call these two regions $R$ and $L$ respectively and the causal development of the right patch is the entanglement wedge of interest.

We now consider a coordinate system $(\rho, \tau, x)$ that covers only the right entanglement wedge [39]. We call these AdS-Rindler coordinates, after the analogous coordinate change in flat space. The explicit coordinate transformation between the global coordinates and $(\rho, \tau, x)$ is

$$
\begin{aligned}
r &= \sqrt{\rho^2 \sinh^2 x + \left( \sqrt{\rho^2 - 1} \cosh \eta \cosh \tau + \rho \cosh x \sinh \eta \right)^2}, \\
t &= \arctan\left( \frac{\sinh \tau \sqrt{\rho^2 - 1}}{\rho \cosh x \cosh \eta + \sqrt{\rho^2 - 1} \cosh \tau \sinh \eta} \right), \\
\varphi &= \arctan\left( \frac{\rho \sinh x}{\sqrt{\rho^2 - 1} \cosh \tau \cosh \eta + \rho \cosh x \sinh \eta} \right).
\end{aligned}
\tag{50}
$$

In these Rindler coordinates, the metric becomes that of the BTZ black brane

$$
ds^2 = -(\rho^2 - 1)d\tau^2 + \frac{d\rho^2}{\rho^2 - 1} + \rho^2 dx^2 \qquad x \in \mathbb{R} \qquad \tau \sim \tau + 2\pi i,
\tag{51}
$$

Here the (Rindler) horizon at $\rho = 1$ is precisely the minimal surface $\gamma_A$. Note that (unlike the BTZ black *hole*) here the range of $x$ is truly infinite. The role of the parameter $\eta \in [0, \infty]$ is to describe different Rindler patches that intersect the boundary on intervals of different sizes. For $\eta = 0$, it covers exactly half of AdS. In general, the relation between $\eta$ and the size of the interval is

$$
\cosh \eta = \frac{1}{\sin \frac{\theta}{2}}.
\tag{52}
$$

We show regions corresponding to different values of $\eta$ in Fig. 3. Note that a set of Rindler coordinates similar to equation (50) covers the left patch.

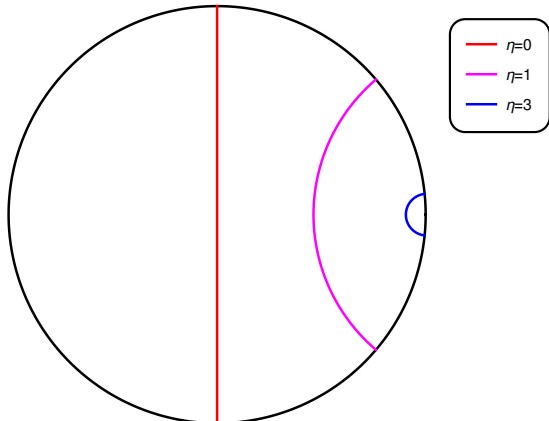

Figure 3: The Rindler regions corresponding to different values of the boost parameter $\eta$.

Such coordinate systems are useful because the vacuum in global coordinates $|0\rangle$ is the thermofield double state when written with respect to Rindler energies, i.e.

$$|0\rangle = \sum_n e^{-\frac{2\pi E_n}{2}} |n^*\rangle_L |n\rangle_R, \tag{53}$$

where $n$ here runs over all energy eigenstates of the full many-body system, and $|n^*\rangle_L$ is defined as the CPT conjugate of the corresponding state on the right side. This will eventually make it easier to compute the entanglement across the two sides. Note that the frequencies in Rindler space are actually continuous making the sum (53) really an integral.

### 3.2.1 Rindler Mode Expansion and Bogoliubov Coefficients

We may globally expand the field in modes that are appropriate to these Rindler coordinates as

$$\phi(\rho, \tau, x) = \sum_{I \in L, R} \int_{\omega > 0} \frac{d\omega}{2\pi} \int \frac{dk}{2\pi} \left( e^{-i\omega\tau} b_{\omega,k,I} g_{\omega,k,I}(\rho, x) + e^{+i\omega\tau} b_{\omega,k,I}^\dagger g_{\omega,k,I}^*(\rho, x) \right). \tag{54}$$

A word about the notation is appropriate. $I$ runs over two discrete values, $L$ and $R$, and the mode functions $b_{L,R}$ have support only in the $L$ and $R$ patches respectively. $k$ is continuous as $x$ is an infinite coordinate, and $\omega$ is continuous as it is conjugate to a time $\tau$ associated with a Killing horizon in the bulk[5]. As usual, we sum only over positive frequencies. For notational convenience we will sometimes write $\int_{\omega > 0} \frac{d\omega}{2\pi} \int \frac{dk}{2\pi} = \sum_{\omega,k}$.
The mode functions $g_{\omega,k,I}(\rho, x)$ can be written in terms of hypergeometric functions; we provide explicit expressions in Appendix A. The creation and annihilation operators are normalized to obey the commutation relations

$$[b_{\omega,k,I}, b_{\omega',k',I'}^\dagger] = (2\pi)^2 \delta(\omega - \omega')\delta(k - k')\delta_{II'}. \tag{55}$$

The thermofield double state (53) results in a thermal state with temperature $T = \frac{1}{2\pi}$ when traced over either the left or right sector; thus the Rindler particles created by $b^\dagger$ feel that

---

[5]Strictly speaking, this is inaccurate: to obtain a finite density of states, one actually needs to impose (e.g. a brick-wall) cutoff near the horizon at $\rho = \rho_c \equiv 1 + \epsilon$. This will discretize the spectrum in $\omega$. As the cutoff is taken to approach the horizon, this spectrum will become arbitrarily finely spaced, and any resulting divergences will be seen to be the usual *bulk* UV divergences of the entanglement entropy. However, our calculation is not sensitive to such UV issues, and we thus treat $\omega$ as continuous from the outset.

they are in a thermal state and all occupation numbers for the $b$ oscillators will be populated according to the Bose distribution.

By acting on the state (53) with creation and annihilation operators, we may also derive the following facts about their action on the vacuum:

$$b_{\omega,k,L}|0\rangle = e^{-\frac{2\pi\omega}{2}}b_{\omega,-k,R}^{\dagger}|0\rangle, \qquad b_{\omega,k,L}^{\dagger}|0\rangle = e^{\frac{2\pi\omega}{2}}b_{\omega,-k,R}|0\rangle. \tag{56}$$

The sign flip of the spatial momentum arises from the CPT conjugation. Note that creating a particle localized outside the entangling region is equivalent to annihilating one *inside* the entangling region: this counter-intuitive effect arises from the high degree of entanglement present in the relativistic vacuum. We would also like to emphasize that the relations (56) are not true as operator statements, but only when acting on the global vacuum.

Now we note that the creation and annihilation operators in the two coordinate systems are related by Bogoliubov coefficients $\alpha, \beta$:

$$a_{m,n} = \sum_{I,\omega,k}\left(\alpha_{m,n;\omega,k,I}b_{\omega k,I} + \beta_{m,n;\omega,k,I}^{*}b_{\omega,k,I}^{\dagger}\right). \tag{57}$$

The commutation relations (55) lead to the following orthogonality constraint on the Bogoliubov coefficients:

$$\sum_{I,\omega,k}\left(\alpha_{m,n;\omega k,I}\alpha_{m'n';\omega,k,I}^{*} - \beta_{m,n;\omega,k,I}^{*}\beta_{m',n';\omega,k,I}\right) = \delta_{m,m'}\delta_{n,n'}. \tag{58}$$

In this work we study only the primary mode (41) with $m = n = 0$; thus we will henceforth omit the $m, n$ indices on the Bogoliubov coefficients, i.e. $\alpha_{\omega,k,I} \equiv \alpha_{0,0;\omega,k,I}$. These particular coefficients satisfy extra constraints arising from the fact that the global annihilation operator $a_{0,0}$ annihilates the vacuum $a_{0,0}|0\rangle = 0$. Using the mode expansion (57) we find the following restrictions on the Bogoliubov coefficients

$$\alpha_{\omega,k,L} = -e^{\frac{2\pi\omega}{2}}\beta_{\omega,-k,R}^{*}, \qquad \beta_{\omega,k,L}^{*} = -e^{-\frac{2\pi\omega}{2}}\alpha_{\omega,-k,R}. \tag{59}$$

Using these, the normalization condition (58) can be written purely in terms of the right patch as

$$\sum_{\omega,k}\left(|\alpha_{\omega k,R}|^{2}(1 - e^{-2\pi\omega}) - |\beta_{\omega k,R}|^{2}(1 - e^{2\pi\omega})\right) = 1. \tag{60}$$

This relation will play an important role in what follows.

Finally, we turn to the description of the excited single-particle state $|\psi\rangle$ (42) in Rindler coordinates. Using equations (56) and (59) we see that the action of the creation operator $a_{0,0}^{\dagger}$ on the vacuum can be rewritten entirely in terms of the the right sector as

$$|\psi\rangle \equiv a_{0,0}^{\dagger}|0\rangle = \sum_{\omega,k}\left((1 - e^{-2\pi\omega})\alpha_{\omega,k,R}^{*}b_{\omega,k,R}^{\dagger} + (1 - e^{2\pi\omega})\beta_{\omega,k,R}b_{\omega,k,R}\right)|0\rangle. \tag{61}$$

We see that adding a single particle to the vacuum in global coordinates can be understood as a particular superposition of creation and annihilation of particles in the Rindler thermal bath. While the act of creating a particle can be localized on the right side, it is not a unitary operation; if it had been unitary on the right side it would not have changed the entanglement entropy.

The representation (61) will be convenient when we trace out the left sector, and we will use it extensively in what follows. From now on, we will work entirely on the right patch: thus in the remainder of this paper we further suppress the $R$ subscript, e.g. $\alpha_{\omega,k} \equiv \alpha_{\omega,k,R}$. We note that a similar technique was used to compute the bulk entanglement entropy of a single-particle state in [40, 41]. Those computations were somewhat simpler due to a judicious choice of modes. We have no such freedom, as we are interested in finding the entanglement entropy in the exact state corresponding to the primary operator.

### 3.2.2 Explicit Form of the Bogoliubov Coefficients

We now turn to the explicit form of the Bogoliubov coefficients. Obtaining these expressions is not trivial. In principle, they can be obtained by computing the Klein-Gordon inner product between the Rindler and global wave-functions; however those integrals are difficult and we were unable to evaluate them in closed form. Instead we map the problem to a boundary integral, as explained in detail in Appendix A.

The result is

$$
\begin{aligned}
\alpha_{\omega,k} &= \frac{1}{(\cosh\eta)^{2h}}\left[\frac{e^\eta - i}{e^\eta + i}\right]^{i\omega} F(\omega, k), \\
\beta_{\omega,k} &= -\frac{1}{(\cosh\eta)^{2h}}\left[\frac{e^\eta + i}{e^\eta - i}\right]^{i\omega} F(\omega, k).
\end{aligned}
\tag{62}
$$

where $F(\omega, k)$ is the following $\eta$-independent function:

$$
F(\omega, k) = \frac{2^{2h}}{\Gamma(2h)}\sqrt{\frac{2\omega\Gamma(i\omega)\Gamma(-i\omega)\Gamma\left(h + i\frac{\omega-k}{2}\right)\Gamma\left(h + i\frac{\omega+k}{2}\right)\Gamma\left(h - i\frac{\omega-k}{2}\right)\Gamma\left(h - i\frac{\omega+k}{2}\right)}{8\pi}}. \tag{63}
$$

As far as we are aware this is the first closed-form presentation of these Bogoliubov coefficients, and we anticipate that they will have further applications.

We now discuss their behavior at large $\eta$, corresponding to small interval sizes. In this limit the entangling region is a very small part of global AdS, which extends from $r \to \infty$ up to $r \sim \alpha^{-1} \sim e^\eta$. In this region the global wavefunction (41) is of order $r^{-2h}$, and we thus expect the overlap with the $\mathcal{O}(1)$ Rindler wavefunction to scale as

$$
\alpha \sim \beta \sim e^{-2\eta h}. \tag{64}
$$

At fixed and finite $\omega$, the explicit expressions above do indeed display this scaling. However the Bogoliubov coefficients cannot uniformly vanish as $\eta \to \infty$, as their integral over all $\omega$ and $k$ is fixed by the normalization condition (60). In particular, note that in the expression (62) for $\beta$, the $\omega$ dependence arising from the bracketed factor is $\exp(-\frac{\omega}{\omega_0})$ where the threshold frequency $\omega_0 \sim \frac{e^\eta}{2}$. Thus as $\eta$ is increased, more and more spectral weight in the normalization integral is pushed out to larger $\omega$ such that the integral remains constant.

## 4 Bulk Modular Hamiltonian

In this section we develop some of the intermediate results that we will require to compute the bulk entanglement entropy. We will do this by studying the relation between bulk and boundary modular Hamiltonians. In particular, in [5] Jafferis, Lewkowycz, Maldacena and Suh (JLMS) postulated that for any state that is a perturbative excitation of the vacuum, and for any choice of boundary spatial region $A$, the following equality holds as a relation between operators on the bulk semi-classical Hilbert space:

$$
K_{\text{CFT}} = K_{\text{bulk}} + \frac{\hat{A}}{4G_N}. \tag{65}
$$

Here $K_{\text{CFT}}$ is the CFT modular Hamiltonian corresponding to a region on the boundary, $\hat{A}$ is the minimal area *operator* which measures the area of the usual minimal surface, and $K_{\text{bulk}}$ is the modular Hamiltonian for perturbative degrees of freedom in the *bulk* associated with the entanglement wedge. In our calculation $K_{\text{bulk}}$ is the modular Hamiltonian of the scalar field

$\phi$.

We note that this expression is state-dependent, in that the $K$ terms are operators that also depend on the state in question. We will sometimes denote this with a superscript, e.g. $K^0$ is the modular Hamiltonian for a reduced density matrix arising from the vacuum, $K^\psi$ is the one for the excited state, etc. The $\hat{A}$ term, on the other hand, is always the same state-independent operator. Note in particular that the entanglement entropy that we are calculating is

$$\langle \psi | K^\psi_{\text{CFT}} | \psi \rangle - \langle 0 | K^0_{\text{CFT}} | 0 \rangle . \tag{66}$$

We leave evaluation of this to the next section: here we instead study the implementation of equation (65) in a slightly simpler context. In particular, we consider the modular Hamiltonian associated with the vacuum but study the difference in its expectation value between the one-particle state and the vacuum, i.e. defining the notation for any fixed operator $X$, $\Delta X \equiv \langle \psi | X | \psi \rangle - \langle 0 | X | 0 \rangle$, we study

$$\Delta K^0_{\text{CFT}} = \Delta K^0_{\text{bulk}} + \frac{\Delta \hat{A}}{4 G_N} . \tag{67}$$

Here the superscript in 0 reminds us that the modular Hamiltonians are those associated with the vacuum. We will compute both sides of this and demonstrate agreement. The intermediate steps in this calculation will provide building blocks for the final calculation to follow.

We begin by computing the left-hand side. The modular Hamiltonian of an interval of length $\theta$ in the CFT vacuum on the cylinder is [42]

$$K^0_{\text{CFT}} = 2\pi \int_0^\theta d\varphi \left( \frac{\cos\left(\varphi - \frac{\theta}{2}\right) - \cos \frac{\theta}{2}}{\sin \frac{\theta}{2}} \right) T_{tt} . \tag{68}$$

Using $2\pi \langle \psi | T_{tt} | \psi \rangle = 2h$, we find immediately that [32]

$$\Delta K^0_{\text{CFT}} = 2h \left( 2 - \theta \cot\left( \frac{\theta}{2} \right) \right) . \tag{69}$$

We have encountered this precise expression before in equation (26), where it arose as the identity contribution to all OPE contractions in the CFT replica trick calculation.

## 4.1 Shift in Minimal Area

We now turn to the gravitational side. We begin with the shift in the minimal area; this arises due to the gravitational backreaction of the particle. As we have emphasized, in general the one-particle state is not a small perturbation of the vacuum; however, in the large $c$ (small $G_N$ limit), the backreaction of the particle on the metric *is* small, and we may thus treat this as a linear perturbation of the classical minimal area. We have

$$\Delta \hat{A} \equiv \langle \psi | \hat{A} | \psi \rangle - \langle 0 | \hat{A} | 0 \rangle = \delta_g A[g_0] + \mathcal{O}(G_N^2), \tag{70}$$

where $A[g]$ denotes the classical minimal area functional, $g_0$ is the metric of empty AdS$_3$, and $\delta_g$ is the perturbation to the metric caused by the backreaction of the bulk particle, as computed in equation (47).

This computation is entirely straightforward. The area of the minimal surface in empty AdS$_3$ reads

$$A[g_0] = 2 \int_{r_{\text{min}}}^\infty dr \sqrt{\frac{1}{r^2 + 1} + r^2 (\varphi'(r))^2} , \tag{71}$$

where $\varphi'(r)$ given by

$$\varphi'(r) = \frac{r_{\min}}{r\sqrt{(r^2 - r_{\min}^2)(1 + r^2)}},$$ (72)

and where $r_{\min}$ is the deepest point that the surface reaches in the bulk. It is related to the size of the interval $\theta$ through

$$\theta = 2\arctan\left[\frac{1}{r_{\min}}\right].$$ (73)

We may now correct this minimal area by taking into account the shift in the metric in equation (47). To leading order the location of the minimal surface does not change. The correction to the area is then simply

$$\delta_g A[g_0] = 2\int_{r=r_{\min}}^{\infty} dr\, 8G_N h\left(1 - \frac{1}{(r^2+1)^{2h-1}}\right)\frac{\sqrt{1 - \frac{r_{\min}^2}{r^2}}}{(1+r^2)^{\frac{3}{2}}},$$ (74)

which gives

$$\frac{\Delta\hat{A}}{4G_N} = 2h\left(2 - \theta\cot\left[\frac{\theta}{2}\right]\right)$$
$$- \frac{\Gamma\left(\frac{3}{2}\right)\Gamma(2h+1)}{\Gamma\left(\frac{3}{2}+2h\right)}\tan\left[\frac{\theta}{2}\right]^{4h}{}_2F_1\left[2h, \frac{1}{2}+2h, \frac{3}{2}+2h; -\tan\left[\frac{\theta}{2}\right]^2\right].$$ (75)

We now pause to discuss the physical significance of these two terms. The first term is precisely equal to the shift in the expectation of the CFT modular Hamiltonian (69). From the bulk point of view it arises from the structure of the asymptotic metric, and to this order we would have obtained precisely the same term even if we had been dealing with a delta-function of stress-energy localized at the center[6].

The second term is less universal; we can see that it is probing the interior of the geometry and so depends on the details of the lump of energy in the center. In a small-$\theta$ expansion it turns out to depend non-analytically on $\theta$ as $\theta^{4h}$, allowing it to be easily distinguished from the analytic dependence on $\theta$ in the first term.

Finally, we can expand this answer in the small interval limit. We find

$$\frac{\Delta\hat{A}}{4G_N} = h\frac{\theta^2}{3}\left(1 + \mathcal{O}(\theta^2)\right) - \left(\frac{\theta}{2}\right)^{4h}\frac{\Gamma\left(\frac{3}{2}\right)\Gamma(1+2h)}{\Gamma\left(\frac{3}{2}+2h\right)}\left(1 + \mathcal{O}(\theta^2)\right) + \dots.$$ (76)

We will make use of this expression in the next subsection.

## 4.2 Bulk Modular Hamiltonian

We now turn to an evaluation of the shift of $K_{\text{bulk}}^0$. This operator is the modular Hamiltonian for the scalar field density matrix associated with the bulk entangling region $\Sigma_A$ in the vacuum. This has a local expression in terms of the scalar field stress tensor. In particular, tracing out the region outside $\gamma_A$ in the thermofield representation of the vacuum state (53), we find that this density matrix takes the form

$$\rho_0 = \sum_n e^{-2\pi E_n}|n\rangle\langle n| \equiv e^{-K_{\text{bulk}}^0},$$ (77)

---

[6]To see this explicitly, one can consider the entanglement entropy in conical defect geometries. The answer for the entanglement entropy in those backgrounds is given for example in (3.2) and (3.3) of [19] (see also [43]). Expanding to first order in $h/c$, we get precisely (69), which was the identity contribution to the CFT correlator. The higher orders in $h/c$ come if one includes the entire Virasoro identity block.

where here $E_n$ is the energy conjugate to the Killing time $\tau$ in the Rindler coordinates (51). We may thus write $K_{\text{bulk}}^0$ as an integral over the scalar field stress tensor $T_{\mu\nu}$ defined in equation (44) as

$$K_{\text{bulk}}^0 = 2\pi \int_{\Sigma_A} \sqrt{|g_{\Sigma_A}|}\, d\Sigma_A \xi^\nu N^\mu T_{\mu\nu}, \tag{78}$$

where $g_{\Sigma_A}$ is the induced metric on the entangling region, $\xi^\nu$ is the Killing vector generating $\tau$ translations, and $N^\mu$ is the normal vector to $\Sigma_A$. We now evaluate $\langle\psi|K_{\text{bulk}}^0|\psi\rangle$, which simply counts the bulk energy inside the entangling region.

To evaluate this integral, it is simplest to use the coordinate transformations (50) to convert the global coordinate expressions for $\langle\psi|T_{\mu\nu}|\psi\rangle$ in equation (46) to Rindler coordinates (51). We may then integrate over all $x \in \mathbb{R}$, $\rho > 1$ at a fixed Rindler time $\tau$. This precisely covers $\Sigma_A$.

The resulting answer is somewhat unwieldy and complicated, but in an expansion in powers of $\theta$ we find the leading term to be

$$\Delta K_{\text{bulk}}^0 = \theta^{4h} 4^{1-2h} h(2h-1) \int_{\Sigma_A} d\rho\, dx\, \rho\left(\rho^2 - 1 + \rho\cosh x\left(2\sqrt{\rho^2-1}+\rho\cosh x\right)\right)^{-2h}. \tag{79}$$

The result of the integral is

$$\Delta K_{\text{bulk}}^0 = \left(\frac{\theta}{2}\right)^{4h} \frac{\Gamma\left(\frac{3}{2}\right)\Gamma(1+2h)}{\Gamma\left(\frac{3}{2}+2h\right)}\left(1 + \mathcal{O}(\theta^2)\right). \tag{80}$$

We have demonstrated this analytically for integer values of $h$ and numerically for all other values. It is precisely equal in magnitude and opposite in sign to the negative term in equation (76). We will use this result in the next sub-section.

Assembling together equations (76) and (80) into the right hand side of equation (67), we see that all non-analytic terms in $\theta$ cancel, and what remains precisely agrees with the (small $\theta$ expansion of) equation (69). In other words, the JLMS formula (67) holds.

We note that this is not a miraculous consequence of holography; rather, the implementation of equation (65) in this particular state (and to leading order in $G_N$) follows from semi-classical gravity [5, 44, 45]. We briefly summarize the argument, whose essence follows from a gravitational Gauss law. More specifically, Wald's treatment of the first law of black hole mechanics [46, 47] tells us that if we have a spacetime with a timelike Killing vector $\zeta$ that degenerates on a horizon $\gamma_A$, then the change in the area $\Delta A$ of the horizon under a small perturbation of the metric $\delta g$ is related to the change in boundary energy $\Delta K_{\text{CFT}}^0$ as

$$\Delta K_{\text{CFT}}^0 = \frac{\Delta A}{4G_N} + \int_{\Sigma_A} d\Sigma_A \xi^\nu N^\mu \delta E_{\mu\nu}[g], \tag{81}$$

where $\delta E_{\mu\nu}$ is the linearization of the geometric part of the Einstein equation, including a cosmological constant term. The above relation holds for any perturbation $\delta g$; however if we now restrict to on-shell perturbations, then the Einstein equations relate $\delta E_{\mu\nu}[g]$ to the scalar field stress tensor, which when combined with equation (78) results immediately in equation (67).

## 5 Bulk Entanglement Entropy

We now turn to the computation of the bulk entanglement entropy with the aim of testing the FLM proposal

$$S_{\text{EE}}^{\text{CFT}} = \frac{A_{\text{min}}}{4G_N} + S_{\text{EE}}^{\text{bulk}}. \tag{82}$$

We computed the quantum corrected minimal area in equation (76) and by comparing with the CFT answer (31), we expect to find

$$\Delta S_{\text{EE}}^{\text{bulk}} \approx \left(\frac{\theta}{2}\right)^{4h} \frac{\Gamma(3/2)\Gamma(2h+1)}{\Gamma(2h+3/2)} - \left(\frac{\theta}{2}\right)^{8h} \frac{\Gamma(3/2)\Gamma(4h+1)}{\Gamma(4h+3/2)}. \tag{83}$$

In this section, we reproduce this result by a bulk computation. The bulk density matrix is given by

$$\rho = a^\dagger \left|0\right\rangle \left\langle 0\right| a. \tag{84}$$

As explained in Section 3.2, it can be rewritten in terms of Rindler modes using the Bogoliubov transformation (57) as

$$\begin{aligned} \rho &= \sum_{\omega,k} \left((1-e^{-2\pi\omega})\alpha_{\omega,k}^* b_{\omega,k}^\dagger + (1-e^{2\pi\omega})\beta_{\omega,k} b_{\omega,k}\right) \left|0\right\rangle\left\langle 0\right| \\ &\times \sum_{\omega',k'} \left((1-e^{-2\pi\omega'})\alpha_{\omega',k'} b_{\omega',k'} + (1-e^{2\pi\omega'})\beta_{\omega',k'}^* b_{\omega',k'}^\dagger\right). \end{aligned} \tag{85}$$

Since all operators act solely on the right wedge, it is easy to trace out the left region to obtain

$$\begin{aligned} \rho_1 \equiv \text{Tr}_{\mathcal{H}_L} \rho &= \sum_{\omega,k} \left((1-e^{-2\pi\omega})\alpha_{\omega,k}^* b_{\omega,k}^\dagger + (1-e^{2\pi\omega})\beta_{\omega,k} b_{\omega,k}\right) e^{-2\pi H} \\ &\times \sum_{\omega',k'} \left((1-e^{-2\pi\omega'})\alpha_{\omega',k'} b_{\omega',k'} + (1-e^{2\pi\omega'})\beta_{\omega',k'}^* b_{\omega',k'}^\dagger\right), \end{aligned} \tag{86}$$

where

$$H = \sum_{\omega,k} \omega b_{\omega,k}^\dagger b_{\omega,k}. \tag{87}$$

With this representation, we can see that the reduced density matrix of the right wedge is some excitation of the thermal state (77), where we note that $K_{\text{bulk}}^0 = 2\pi H$,

$$\rho_0 = e^{-2\pi H}. \tag{88}$$

Since a direct diagonalization of $\rho_1$ is too difficult, we will proceed with the replica trick. We start by computing the change in the Rényi entropies

$$\Delta S_n^{\text{bulk}} = \frac{1}{1-n} \log \frac{\text{Tr}\rho_1^n}{\text{Tr}\rho_0^n}, \tag{89}$$

before analytically continuing and taking the $n \to 1$ limit. Computing the Rényi entropies is in principle straightforward. All expectation values of the creation and annihilation operators are computed using Wick's theorem for free fields and the thermal two-point functions are given by

$$\text{Tr}\rho_0 b_{\omega,k}^\dagger b_{\omega',k'} = \frac{4\pi^2 \delta(\omega-\omega')\delta(k-k')}{e^{2\pi\omega} - 1}. \tag{90}$$

Given the exact expressions for the Bogoliubov coefficients (62), to compute the $n$-th Rényi entropy we simply need to perform $2n$ integrals over frequencies and momenta, which at the very least can be done numerically. However, we are left with the problem of performing the analytic continuation. Note that this is exactly the same problem as in the CFT: we could compute the Rényi entropies using Wick contractions for generalized free fields, but we were not able to perform the analytic continuation for general conformal dimension. For this reason, we will now consider a small interval limit in which we can match bulk and boundary computations.

## 5.1 The Small Interval Limit

In the CFT, the small interval limit is defined by

$$\theta \ll 1, \tag{91}$$

for $\theta$ the size of the boundary interval. In the bulk, using equation (52), it can be defined as the limit of large boost parameter, namely

$$\eta \gg 1. \tag{92}$$

The boost parameter $\eta$ is a purely kinematical component of the problem and enters the bulk computation only through the Bogoliubov coefficients. One might naively guess that large $\eta$ translates into the Bogoliubov coefficients being small. As explained in Section 3.2.2, this is true pointwise in frequency but does not necessarily hold for integrated quantities; one obvious exception is given by the normalization integral (60) over all Bogoliubov coefficients, which always evaluates to 1. For this reason, care must be taken in expanding the Rényi entropies (89) to a given order in the Bogoliubov coefficients. We need to expand in the true small parameter in the problem which can be isolated by defining

$$\delta\rho \equiv \rho_1 - \rho_0. \tag{93}$$

We know that the matrix elements of $\delta\rho$ must become small in the small interval limit since one should not be able to distinguish between $\rho_1$ and $\rho_0$ in the limit where the right wedge goes to zero size. This will be correct if the two density matrices are close. We will now expand the Rényi entropies order by order in $\delta\rho$ which will enable us to perform the analytic continuation and compute the entanglement entropy.

## 5.2 The First Order Result

We begin with the following expansion

$$
\begin{aligned}
\mathrm{Tr}\rho_1^n &= \mathrm{Tr}(\rho_0 + \delta\rho)^n \\
&= \mathrm{Tr}\rho_0^n + n\mathrm{Tr}\rho_0^{n-1}\delta\rho + \mathcal{O}(\delta\rho^2) \\
&= \mathrm{Tr}\rho_0^n + n\mathrm{Tr}\rho_0^{n-1}\rho_1 - n\mathrm{Tr}\rho_0^n.
\end{aligned}
\tag{94}
$$

From this, we can compute the change in entanglement entropy as

$$\Delta S_{\mathrm{EE}}^{\mathrm{bulk}} = \lim_{n\to 1}\Delta S_n^{\mathrm{bulk}} = -\partial_n \Delta S_n^{\mathrm{bulk}}\big|_{n=1}. \tag{95}$$

The first term in equation (94) is canceled by the denominator of equation (89), coming from the subtraction of the vacuum entanglement entropy. Using the fact that $\mathrm{Tr}\rho_0 = \mathrm{Tr}\rho_1 = 1$, we find

$$\Delta S_{\mathrm{EE}}^{\mathrm{bulk}} = -\mathrm{Tr}\rho_1 \log\rho_0 + \mathrm{Tr}\rho_0 \log\rho_0 + \mathcal{O}(\delta\rho^2). \tag{96}$$

This can be rewritten as

$$\Delta S_{\mathrm{EE}}^{\mathrm{bulk}} = 2\pi\Delta\langle H\rangle + \mathcal{O}(\delta\rho^2), \tag{97}$$

which is nothing else than the first law of entanglement for the bulk quantum field theory. We have already computed the expectation value of this Hamiltonian in position space in equation (80). To build technology for the next section we do so again using momentum space techniques. We have

$$
\begin{aligned}
2\pi\Delta\langle H\rangle &= \mathrm{Tr}\rho_0\Big[ \sum_{\substack{\omega_1,\omega_2,\omega_3 \\ k_1,k_2,k_3}} \Big((1-e^{-2\pi\omega_1})\alpha_{\omega_1,k_1}b_{\omega_1,k_1} + (1-e^{2\pi\omega_1})\beta^*_{\omega_1,k_1}b^\dagger_{\omega_1,k_1}\Big) \\
&\quad \times \Big(2\pi\omega_2 b^\dagger_{\omega_2,k_2} b_{\omega_2,k_2}\Big)\Big((1-e^{-2\pi\omega_3})\alpha^*_{\omega_3,k_3}b^\dagger_{\omega_3,k_3} + (1-e^{2\pi_3\omega_3})\beta_{\omega_3,k_3}b_{\omega_3,k_3}\Big)\Big],
\end{aligned}
\tag{98}
$$

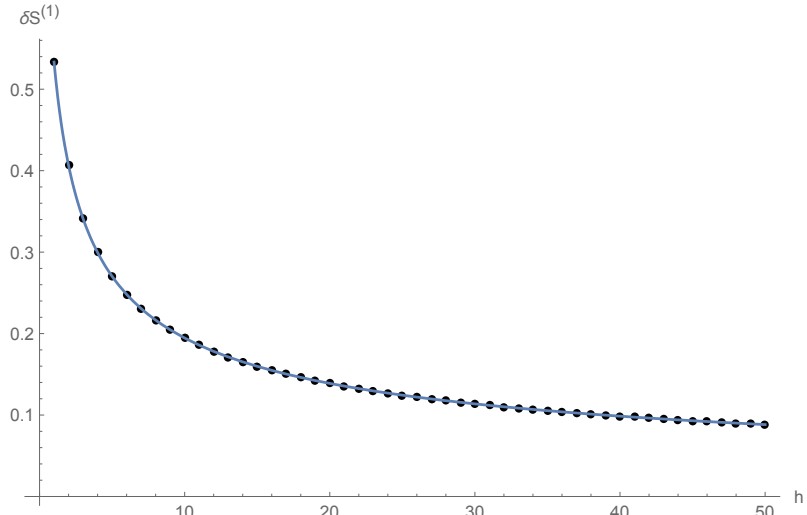

Figure 4: The black dots represent the coefficient of $\theta^{4h}$ in the first order change to the entanglement entropy. The blue line represents the absolute value of the $\theta^{4h}$ coefficient appearing in the area operator, which is of opposite sign. The matching is exact and thus they cancel.

where the subtraction of $\text{Tr}\rho_0 \log \rho_0$ implies that in the above expression we do not allow any Wick contractions between the $b^\dagger_{\omega_2,k_2}$ and $b_{\omega_2,k_2}$.

Evaluating the expectation values using equation (90), we find

$$2\pi\Delta\langle H\rangle = \sum_{\omega,k} 2\pi\omega(|\alpha_{\omega,k}|^2 + |\beta_{\omega,k}|^2). \tag{99}$$

It is important to note that this integral is more convergent at large $\omega$ than the normalization integral (60). We can therefore strip out the factor $(\cosh\eta)^{-4h}$ in the squared Bogoliubov coefficients, set $\eta = \infty$ in the remainder, and still have a finite integral. Plugging in the expression for the Bogoliubov coefficients, we find

$$2\pi\Delta\langle H\rangle = \frac{1}{(\cosh\eta)^{4h}}\int_{\omega>0} d\omega \int dk \frac{2^{4h}\omega^2}{4\pi^2}F_1(\omega,k), \tag{100}$$

with

$$F_1(\omega,k) = \frac{\Gamma(i\omega)\Gamma(-i\omega)\Gamma(h+i\frac{\omega+k}{2})\Gamma(h-i\frac{\omega+k}{2})\Gamma(h+i\frac{\omega-k}{2})\Gamma(h-i\frac{\omega-k}{2})}{(\Gamma(2h))^2}. \tag{101}$$

The double integral can be evaluated numerically and we plot the result in Fig. 4. The prefactor gives a factor $\theta^{4h}$ and we find perfect agreement between the values of the integral and the first term in equation (83).

## 5.3 The Second Order Result

We now turn to the second order contribution. At second order in $\delta\rho$ we have

$$
\begin{aligned}
\text{Tr}\rho_1^n\Big|_{\mathcal{O}(\delta\rho^2)} &= \frac{n}{2}\sum_{m=0}^{n-2}\text{Tr}\delta\rho\rho_0^m\delta\rho\rho_0^{n-2-m} \\
&= \frac{n}{2}\text{Tr}\rho_1\tilde{\rho}(n)\rho_0^{n-2} - n(n-1)\text{Tr}\rho_0^{n-1}\rho_1 + \frac{n(n-1)}{2}\text{Tr}\rho_0^n,
\end{aligned} \tag{102}
$$

where we have defined

$$
\begin{aligned}
\tilde{\rho}(n) &= \sum_{m=0}^{n-2} \rho_0^m \rho_1 \rho_0^{-m} \\
&= \sum_{\substack{\omega_1,\omega_2 \\ k_1,k_2}} \Big[ \frac{1-e^{2\pi(\omega_2-\omega_1)(n-1)}}{1-e^{2\pi(\omega_2-\omega_1)}}(1-e^{-2\pi\omega_1})(1-e^{-2\pi\omega_2})\alpha_1^*\alpha_2 b_1^\dagger \rho_0 b_2 \\
&\quad + \frac{1-e^{-2\pi(\omega_1+\omega_2)(n-1)}}{1-e^{-2\pi(\omega_1+\omega_2)}}(1-e^{-2\pi\omega_1})(1-e^{2\pi\omega_2})\alpha_1^*\beta_2^* b_1^\dagger \rho_0 b_2^\dagger \\
&\quad + \frac{1-e^{2\pi(\omega_1+\omega_2)(n-1)}}{1-e^{2\pi(\omega_1+\omega_2)}}(1-e^{2\pi\omega_1})(1-e^{-2\pi\omega_2})\beta_1\alpha_2 b_1 \rho_0 b_2 \\
&\quad + \frac{1-e^{2\pi(\omega_1-\omega_2)(n-1)}}{1-e^{2\pi(\omega_1-\omega_2)}}(1-e^{2\pi\omega_1})(1-e^{2\pi\omega_2})\beta_1\beta_2^* b_1 \rho_0 b_2^\dagger \Big].
\end{aligned}
\tag{103}
$$

From now on we will work with the shortened notation $\alpha_1 \equiv \alpha_{\omega_1,k_1}$ and identically for the $b, b^\dagger, \beta$. We may now take the derivate with respect to $n$ and we obtain

$$
\Delta S_{\text{EE}}^{(2)} = \frac{1}{2} - \frac{1}{2}\text{Tr}\rho_1 \tilde{\rho}'(1)\rho_0^{-1},
\tag{104}
$$

where we used the fact that $\tilde{\rho}(1) = 0$. In total this gives

$$
\begin{aligned}
\Delta S_{\text{EE}}^{(2)} = \frac{1}{2} + \frac{1}{2} \sum_{\substack{\omega_1,\omega_2,\omega_3,\omega_4 \\ k_1,k_2,k_3,k_4}} \Big[ \\
2\pi(\omega_2-\omega_1)\frac{(1-e^{-2\pi\omega_1})(e^{2\pi\omega_2}-1)(1-e^{-2\pi\omega_3})(1-e^{-2\pi\omega_4})}{1-e^{2\pi(\omega_2-\omega_1)}}\alpha_1^*\alpha_2\alpha_3^*\alpha_4\text{Tr}\rho_0 b_4 b_1^\dagger b_2 b_3^\dagger \\
+ 2\pi(\omega_2-\omega_1)\frac{(1-e^{-2\pi\omega_1})(e^{2\pi\omega_2}-1)(1-e^{-2\pi\omega_3})(1-e^{-2\pi\omega_4})}{1-e^{2\pi(\omega_2-\omega_1)}}\alpha_1^*\alpha_2\beta_3\beta_4^*\text{Tr}\rho_0 b_4^\dagger b_1^\dagger b_2 b_3 \\
+ 2\pi(\omega_1-\omega_2)\frac{(1-e^{2\pi\omega_1})(e^{-2\pi\omega_2}-1)(1-e^{2\pi\omega_3})(1-e^{2\pi\omega_4})}{1-e^{2\pi(\omega_1-\omega_2)}}\beta_1\beta_2^*\alpha_3^*\alpha_4\text{Tr}\rho_0 b_4 b_1 b_2^\dagger b_3^\dagger \\
+ 2\pi(\omega_1-\omega_2)\frac{(1-e^{2\pi\omega_1})(e^{-2\pi\omega_2}-1)(1-e^{2\pi\omega_3})(1-e^{2\pi\omega_4})}{1-e^{2\pi(\omega_1-\omega_2)}}\beta_1\beta_2^*\beta_3\beta_4^*\text{Tr}\rho_0 b_4^\dagger b_1 b_2^\dagger b_3 \\
- 2\pi(\omega_1+\omega_2)\frac{(1-e^{-2\pi\omega_1})(e^{-2\pi\omega_2}-1)(1-e^{2\pi\omega_3})(1-e^{-2\pi\omega_4})}{1-e^{-2\pi(\omega_1+\omega_2)}}\alpha_1^*\beta_2^*\beta_3\alpha_4\text{Tr}\rho_0 b_4 b_1^\dagger b_2^\dagger b_3 \\
+ 2\pi(\omega_1+\omega_2)\frac{(1-e^{2\pi\omega_1})(e^{2\pi\omega_2}-1)(1-e^{-2\pi\omega_3})(1-e^{2\pi\omega_4})}{1-e^{2\pi(\omega_1+\omega_2)}}\beta_1\alpha_2\alpha_3^*\beta_4^*\text{Tr}\rho_0 b_4^\dagger b_1 b_2 b_3^\dagger \Big].
\end{aligned}
\tag{105}
$$

Each of these terms has two possible Wick contractions giving a total of twelve terms that contribute. The four terms where the Wick contraction is between the oscillators one and two combine to give exactly $-1/2$, cancelling against that term on the first line. The remaining eight terms give convergent integrals upon factoring out the factor $(\cosh\eta)^{-8h}$ and setting $\eta = 0$. After some math, we find

$$
\Delta S_{\text{EE}}^{(2)} = \frac{1}{(\cosh\eta)^{8h}}\int_{\omega_1>0}d\omega_1 \int_{\omega_2>0}d\omega_2 \int dk_1 \int dk_2 \frac{2^{8h}\omega_1\omega_2}{64\pi^5}F_2(\omega_1,k_1,\omega_2,k_2),
\tag{106}
$$

with

$$
\begin{aligned}
F_2(\omega_1,k_1,\omega_2,k_2) &= (\omega_1-\omega_2)\frac{(1-e^{-2\pi\omega_1})(1-e^{2\pi\omega_2})}{1-e^{2\pi(\omega_2-\omega_1)}}F_1(\omega_1,k_1)F_1(\omega_2,k_2) \\
&\quad + (\omega_1+\omega_2)\frac{(1-e^{2\pi\omega_1})(1-e^{2\pi\omega_2})}{1-e^{2\pi(\omega_1+\omega_2)}}F_1(\omega_1,k_1)F_1(\omega_2,k_2).
\end{aligned}
\tag{107}
$$

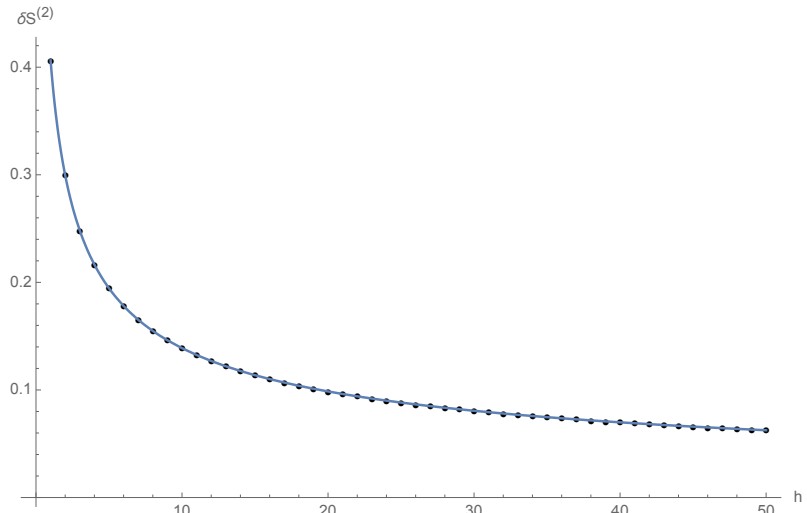

Figure 5: The black dots represent the absolute value of the coefficient of $\theta^{8h}$ coming from the second order change to the entanglement entropy. The blue line represents the absolute value of the $\theta^{8h}$ coefficient as calculated from the CFT and exhibits a perfect match.

This integral can be done numerically and we plot the results in Fig. 5. We find perfect agreement between our results and the second order term in (83). This completes our calculation of the bulk entanglement entropy; we have verified that the FLM formula agrees with the CFT result. We now turn to the discussion and outlook.

# 6 Conclusion and Outlook

## 6.1 Conclusion

In this paper, we successfully tested the FLM and JLMS formulae for quantum corrections to holographic entanglement entropy. We considered CFT states that are obtained by acting on the vacuum by scalar primary single-trace operators and computed their entanglement entropy, in the limit of small subsystem sizes. We then compared this result to a gravity computation. We studied one-particle states in the bulk scalar free field theory, for which we computed the change in the minimal area as well as the bulk entanglement entropy. We found a perfect match between the two[7]

$$
\begin{aligned}
\Delta S_{\text{EE}}^{\text{CFT}} &= \frac{\Delta A_{\min}}{4 G_N} + \Delta S_{\text{EE}}^{\text{bulk}} \\
&= 2h\left(2 - \theta \cot\left(\frac{\theta}{2}\right)\right) - \left(\sin\frac{\theta}{2}\right)^{8h} \frac{\Gamma\left(\frac{3}{2}\right)\Gamma(4h+1)}{\Gamma\left(4h+\frac{3}{2}\right)}(1 + \mathcal{O}(\theta^2)) + \mathcal{O}(\theta^{12h}).
\end{aligned}
\tag{108}
$$

In order to compute this quantity in the bulk scalar field theory, we needed to know the exact form of the Bogoliubov coefficients between global and Rindler $AdS_3$. We provided what we believe is the first calculation of these coefficients in explicit form. The result (108) included a term that appeared both in the $\Delta A_{\min}$ and $\Delta S_{\text{EE}}^{\text{bulk}}$, but with opposite signs. This precise

---

[7]This expression is the full expression that we were able to compute and successfully compare, including the order of all expected corrections. Note however that it is a slight abuse of the small $\theta$ expansion since the first term contains arbitrary high powers of $\theta$. In the strict perturbative expansion, the relevant expression is equation (4).

cancellation, although in general guaranteed by the gravitational Gauss law, provided a nice check of the first law of entanglement for the bulk quantum field theory.

We now discuss various ideas that would provide extensions of our results but still require further investigation.

## 6.2 Outlook

**Finite interval sizes**

First, it would be very interesting to push our results beyond the limit of small intervals. The main barrier for this is the difficulty in the analytic continuation of the Rényi entropies to $n = 1$. In the CFT, the problem seems tractable as one has an explicit closed form expression for any given Rényi entropy $S_n$. It would be interesting to see if one could find a generalization of the Hafnian/Determinant method developed in [28–31] to conformal dimensions different from one.

In the bulk, the problem is slightly harder, since we do not have a closed-form expression for the bulk Rényi entropies. We can express the bulk Rényi entropies $S_n^{\text{bulk}}$ as $2n$ integrals over momentum and frequency, with expressions that are not beyond hope. The main problem is that the integrals do not factorize into $2n$ independent integrals. It would be particularly interesting to see if one could perform the integrals for the case of $h = 1/2$, since in that case we know the analytic continuation exactly in the CFT. We were unfortunately not capable of doing that, so we leave it for future work.

Nevertheless, we would like to emphasize that the only difficulty is in the analytic continuation. This should be contrasted with the case of the mutual information of two spherical regions in higher dimensions [10], where it is not even really known how to proceed beyond the limit of small interval size. Even in three dimensions, the method described in [12] is very hard to deal with on a practical level for arbitrary Rényi entropies.

**Gravitons and photons**

Next, it would be very interesting to consider excitations by operators that are not scalars. In particular, it would be very interesting to take the single-trace operator to be the stress-tensor $T$. In this case, we would be computing the entanglement entropy of boundary gravitons in the bulk. A proper understanding of the entanglement entropy of gravitons is somewhat daunting, combining as it does the usual ambiguities with the factorization of the Hilbert space in a gauge theory [48–52] with new wrinkles associated with the fact that the gauge symmetry in question is diffeomorphism invariance [5,53]. Nevertheless, progress has been made towards dealing with such ambiguities, and it would be very nice to tackle this hard problem in AdS$_3$ where gravitons do not propagate and are purely boundary modes that nevertheless will contribute in a (presumably) calculable manner to the entanglement entropy. In particular, on the CFT side this problem maps to computing the entanglement entropy of Virasoro descendants of the vacuum. This answer is completely determined by Virasoro symmetry since the Rényi entropies are $2n$-point functions of stress tensors. These are completely fixed by the conformal ward identities [54]. The result can thus only depend on the central charge $c$ (this would be drastically different in higher dimensions).

It would also be interesting to consider $U(1)$ currents. In the bulk, this would require computing the entanglement entropy of photons in the one-photon state. Note however that the computation should be quite tractable since the dynamics of the gauge field are given by a Chern-Simons terms and hence are topological. It is particularly interesting to compare the result with the CFT since this is precisely the case $h + \bar{h} = 1$, where the analytic continuation is feasible! We hope to return to these questions in future work.

**Quantum corrections to the Rényi entropies**

It would also be interesting to compute the quantum corrections to the CFT Rényi entropies in the bulk directly. At the leading order, this can be done by introducing a cosmic brane in the bulk [55]. The quantum corrections to this formula were discussed in [4] and one expects[8]

$$n^2 \partial_n \left( \frac{n-1}{n} S_n^{\text{CFT}} \right) = \frac{A_{\text{CB}}}{4G_N} + n^2 \partial_n \left( \frac{n-1}{n} S_n^{\text{bulk}} \right). \tag{109}$$

to hold. To check this, one would need to understand the nature of the interaction between the matter and cosmic brane backreactions. It would be very interesting to try to work out these details in our simple setup. Note that there is a $n$-derivative acting on the Rényi entropy which means we still need to have analytic control over the boundary and bulk Rényi entropies to compute both sides of this formula.

**Higher dimensions**

Finally, it would be interesting to extend our results to dimensions $d > 2$. Many things would go through smoothly, one can still consider one-particle states in the bulk, still compute Bogoliubov coefficients since they are fixed by symmetry and much of the analysis would go through in a similar fashion. In the CFT however, things become more complicated. In $d = 2$, the Rényi entropies are given by $2n$-point functions in the vacuum, which are basically fixed with the generalized free field assumption. For $d > 2$, the $2n$-point function is not on flat-space but rather on $S^1 \times H_{d-1}$ [56], where the periodicity of the spatial circle is $2\pi n$. Even with the factorization assumption, one is left with two-point functions at finite temperature on hyperbolic space. These are not universally fixed by the conformal dimension of the operator.

In fact, even the Rényi entropies themselves are no longer completely fixed by the central charge. For example, they could drastically differ if the CFT is strongly coupled or weakly coupled. The hyperbolic black holes of [57] can only be trusted if the gravity dual is Einstein gravity. The situation is in fact even more complicated, since even for Einstein duals new phases corresponding to hairy hyperbolic black holes can exist [58,59].

This should be contrasted with $d = 2$ where 3d dual gravity is in some sense universal. Because of Virasoro symmetry, the coupling of matter to gravity is universal in AdS$_3$. It is worth noting that the assumption used for the calculations in this paper only required large $c$ factorization of the matter and never demanded the bulk dual to be Einstein. In fact, if one had considered the CFT to be the D1-D5 system at the orbifold point (zero coupling), our results still apply! This is true even though at the orbifold point the bulk dual is full-blown string theory. In that setting, there is in principle no reason to suspect that the RT formula is accurate. This really highlights the pecularity of 3d gravity where many features are perhaps more universal than they "should be" [17,60,61].

To return to higher dimensions, the implication of these facts is that one could still perform computations in the small subsystem limit, but only there. Away from that point, one would need strong additional assumptions and direct input coming from the bulk, with no natural CFT justification. It would be interesting to pursue these ideas further.

# Acknowledgements

We are happy to thank Cesar Agon, Alejandra Castro, Tom Faulkner, Ben Freivogel, Nima Lashkari, Aitor Lewkowycz, Alex Maloney and especially Simon Ross, Gabor Sarosi and

---

[8]We are grateful to Aitor Lewkowycz for pointing this out to us.

Tomonori Ugajin for helpful discussions. The inspiration for this work arose partly from discussions at the D-ITP Entanglement Workshop at the University of Amsterdam. This work was partly completed at the Aspen Center for Physics, which is supported by National Science Foundation grant PHY-1607761. AB would like to thank the String Theory group at ETH for hospitality. AB is supported by the NWO VENI grant 680-47-464 / 4114. NI would like to thank Delta ITP at the University of Amsterdam for hospitality. NI is supported in part by the STFC under consolidated grant ST/L000407/1. SFL would like to acknowledge support from the Netherlands Organisation for Scientific Research (NWO) which is funded by the Dutch Ministry of Education, Culture and Science (OCW).

# A   Computation of the Bogoliubov Coefficients

## A.1   Rindler Mode Functions

Recall that the bulk scalar field can be expanded in terms of the Rindler modes as follows

$$\phi(\rho,\tau,x) = \sum_{I \in L,R} \int_{\omega > 0} \frac{d\omega}{2\pi} \int \frac{dk}{2\pi} \left( e^{-i\omega\tau} b_{\omega,k,I} g_{\omega,k,I}(\rho,x) + e^{+i\omega\tau} b^{\dagger}_{\omega,k,I} g^{*}_{\omega,k,I}(\rho,x) \right), \quad (110)$$

with the left and the right Rindler mode functions $g_{\omega,k,I}$ given by [62]

$$g_{\omega,k,I} = N_{\omega,k} \, e^{ikx} \rho^{-2h} \left(1 - \frac{1}{\rho^2}\right)^{-\frac{i\omega}{2}} {}_2F_1\left[h - \frac{i(\omega+k)}{2}, h - \frac{i(\omega-k)}{2}, 2h; \frac{1}{\rho^2}\right]. \quad (111)$$

In the above, there are two pairs of coordinate systems $(\rho,\tau,x)$, one each for left and right wedges. The number $N_{\omega,k}$ is the normalization constant is given in [8]

$$N_{\omega,k} = \sqrt{\frac{\Gamma\left(h + \frac{i(\omega-k)}{2}\right)\Gamma\left(h + \frac{i(\omega+k)}{2}\right)\Gamma\left(h - \frac{i(\omega-k)}{2}\right)\Gamma\left(h - \frac{i(\omega+k)}{2}\right)}{2\omega\,\Gamma(2h)^2\Gamma(i\omega)\Gamma(-i\omega)}}, \quad (112)$$

where we have taken $N_{\omega,k}$ to be real. In the limit $\rho \to \infty$, the mode functions satisfy

$$\lim_{\rho \to \infty} \rho^{2h} g_{\omega,k,I}(\rho,x) = e^{ikx} N_{\omega,k}. \quad (113)$$

## A.2   Bogoliubov Coefficients

As described in section 3.2.1, Bogoliubov coefficients relate the creation and annihilation operators of the global modes to those in the right and the left Rindler patches. Since the creation operator in global coordinates is a linear combination of these, we have the relation (61)

$$a^{\dagger}_{0,0}|0\rangle = \sum_{\omega,k} \left( (1 - e^{-2\pi\omega})\alpha^{*}_{\omega,k,R} b^{\dagger}_{\omega,k,R} + (1 - e^{2\pi\omega})\beta_{\omega,k,R} b_{\omega,k,R} \right)|0\rangle. \quad (114)$$

We now calculate the Bogoliubov coefficients explicitly. We start by looking at the two-point function

$$\lim_{r \to \infty} r^{2h} \langle 0| \phi(r,t,\varphi) a^{\dagger}_{0,0} |0\rangle, \quad (115)$$

where $\phi(r,t,\varphi)$ is the field in the global coordinates. Using (41), we get

$$\lim_{r \to \infty} r^{2h} \langle 0| \phi(r,t,\varphi) a^{\dagger}_{0,0} |0\rangle = \frac{e^{-2iht}}{\sqrt{2\pi}}. \quad (116)$$

Let us now rewrite the above expression using Rindler coordinates and modes. Substituting the expressions for $\phi$ and $a^\dagger$ in terms of right Rindler modes in equation (116), we obtain

$$
\frac{e^{-2iht(\tau,x)}}{\sqrt{2\pi}} = \lim_{\rho\to\infty} r(\rho,\tau,x)^{2h}\, \langle 0|\sum_{\omega,k}\left(e^{-i\omega\tau}\,g_{\omega k}(\rho,x)\,b_{\omega k} + e^{i\omega\tau}\,g^*_{\omega k}(\rho,x)\,b^\dagger_{\omega k}\right)
$$
$$
\times \sum_{\omega',k'}\left((1-e^{-2\pi\omega'})\,\alpha^*_{\omega',k'}\,b^\dagger_{\omega',k'} + (1-e^{2\pi\omega'})\,\beta_{\omega',k'}\,b_{\omega',k'}\right)|0\rangle\,,
\tag{117}
$$

where we have omitted the subscript $R$ and used the shorthand notation

$$
\sum_{\omega,k} \equiv \int\frac{d\omega}{2\pi}\int\frac{dk}{2\pi}\,.
\tag{118}
$$

Using the following standard correlators,

$$
\langle 0|\,b_{\omega,k}\,b_{\omega',k'}\,|0\rangle = 0\,, \qquad \langle 0|\,b^\dagger_{\omega,k}\,b^\dagger_{\omega',k'}\,|0\rangle = 0\,,
\tag{119}
$$

$$
\langle 0|\,b^\dagger_{\omega,k}\,b_{\omega',k'}\,|0\rangle = \frac{(2\pi)^2}{(e^{2\pi\omega}-1)}\delta(\omega-\omega')\,\delta(k-k')\,,
\tag{120}
$$

$$
\langle 0|\,b_{\omega,k}\,b^\dagger_{\omega',k'}\,|0\rangle = \frac{(2\pi)^2\,e^{2\pi\omega}}{(e^{2\pi\omega}-1)}\delta(\omega-\omega')\,\delta(k-k')\,,
\tag{121}
$$

we can simplify equation (117) to obtain.

$$
\frac{e^{-2iht(\tau,x)}}{\sqrt{2\pi}} = \lim_{\rho\to\infty} r(\rho,\tau,x)^{2h}\sum_{\omega,k}\left[e^{-i\omega\tau}\,g_{\omega,k}(\rho,x)\,\alpha^*_{\omega k} - e^{i\omega,\tau}\,g^*_{\omega,k}(\rho,x)\,\beta_{\omega,k}\right].
\tag{122}
$$

Using equation 50, in the limit of large $\rho$, we get

$$
\lim_{\rho\to\infty} t = \arctan\left(\frac{\sinh\tau}{\cosh x\cosh\eta + \cosh\tau\sinh\eta}\right) \equiv t(\tau,x)\,,
\tag{123}
$$

$$
\lim_{\rho\to\infty} r(\rho,\tau,x)^{2h} = \lim_{\rho\to\infty}\rho^{2h}\left(\sinh^2 x + (\cosh\eta\cosh\tau + \sinh\eta\cosh x)^2\right)^h.
\tag{124}
$$

Also, from equation (113) we know that

$$
\lim_{\rho\to\infty}\rho^{2h}\,g_{\omega,k,I}(\rho,x) = e^{ikx}\,N_{\omega,k}\,,
\tag{125}
$$

Therefore, equation (122) simplifies to

$$
\frac{e^{-2iht(\tau,x)}}{\sqrt{2\pi}} = \left(\sinh^2 x + (\cosh\eta\cosh\tau + \sinh\eta\cosh x)^2\right)^h
$$
$$
\times \sum_{\omega,k}\left[e^{-i\omega\tau+ikx}\,N_{\omega,k}\,\alpha^*_{\omega,k} - e^{i\omega\tau-ikx}\,N^*_{\omega,k}\,\beta_{\omega,k}\right].
\tag{126}
$$

Collecting all functions of $(\tau,x)$ on one side, we define

$$
\mathscr{B}(\tau,x) \equiv \frac{e^{-2iht(\tau,x)}}{\sqrt{2\pi}\left(\sinh^2 x + (\cosh\eta\cosh\tau + \sinh\eta\cosh x)^2\right)^h}\,,
\tag{127}
$$

to obtain

$$
\int\frac{d\omega dk}{(2\pi)^2}\left[e^{-i\omega\tau+ikx}\,N_{\omega,k}\,\alpha^*_{\omega,k} - e^{i\omega\tau-ikx}\,N^*_{\omega,k}\,\beta_{\omega,k}\right] = \mathscr{B}(\tau,x)\,.
\tag{128}
$$

One can then take a Fourier transform of both sides above to show that

$$\alpha_{\omega,k} = \frac{1}{N_{\omega,k}^*} \int\limits_{-\infty}^{\infty} d\tau\, dx\, e^{-i\omega\tau+ikx}\, \mathscr{B}^*(x,t), \tag{129}$$

$$\beta_{\omega,k} = -\frac{1}{N_{\omega,k}^*} \int\limits_{-\infty}^{\infty} d\tau\, dx\, e^{-i\omega\tau+ikx}\, \mathscr{B}(x,\tau). \tag{130}$$

To calculate these integrals, it is easiest to work in light-cone coordinates defined by

$$x^+ \equiv \frac{x+\tau}{2}, \qquad x^- \equiv \frac{x-\tau}{2}. \tag{131}$$

In these coordinates the function $\mathscr{B}(\tau,x)$ factorizes to give

$$\mathscr{B}(x^+,x^-) = \frac{2^{4h}e^{2\eta h}}{\sqrt{2\pi}}\, \mathscr{B}^+(x^+)\cdot \mathscr{B}^-(x^-). \tag{132}$$

Let us focus on the Bogoliubov coefficient $\alpha_{\omega,k}$ for now. Here we need to be careful to take the complex conjugate of $\mathscr{B}(x^+,x^-)$. After some simplifications, we find that

$$\mathscr{B}^{*+}(x^+) \equiv \left(\frac{e^{2x^+}}{(e^\eta + e^{\eta+2x^+} - ie^{2x^+} + i)^2}\right)^h, \tag{133}$$

$$\mathscr{B}^{*-}(x^-) \equiv \left(\frac{e^{2x^-}}{(e^\eta + e^{\eta+2x^-} + ie^{2x^-} - i)^2}\right)^h. \tag{134}$$

The Bogoliubov coefficient then becomes

$$\alpha_{\omega,k} = \frac{2}{N_{\omega,k}^*}\frac{2^{4h}e^{2\eta h}}{\sqrt{2\pi}}\left[, \int\limits_{-\infty}^{\infty} dx^+\, e^{i(k-\omega)x^+}\, \mathscr{B}^{*+}(x^+)\right]\left[\int\limits_{-\infty}^{\infty} dx^-\, e^{i(k+\omega)x^-}\, \mathscr{B}^{*-}(x^-)\right], \tag{135}$$

where the factor of 2 comes from the Jacobian. The integrals can now be computed analytically and read

$$\int\limits_{-\infty}^{\infty} dx^+\, e^{i(k-\omega)x^+}\, \mathscr{B}^{*+}(x^+) = (e^\eta + i)^{-2h}\left(\frac{1+ie^\eta}{-1+ie^\eta}\right)^{-h-\frac{i(k-\omega)}{2}}\frac{1}{2\Gamma(2h)}$$
$$\times \Gamma\left[h+\frac{i(k-\omega)}{2}\right]\Gamma\left[h-\frac{i(k-\omega)}{2}\right]. \tag{136}$$

Similarly we can show that

$$\int\limits_{-\infty}^{\infty} dx^-\, e^{i(k+\omega)x^-}\, \mathscr{B}^{*-}(x^-) = (e^\eta - i)^{-2h}\left(\frac{-1+ie^\eta}{1+ie^\eta}\right)^{-h-\frac{i(k+\omega)}{2}}\frac{1}{2\Gamma(2h)}$$
$$\times \Gamma\left[h+\frac{i(k+\omega)}{2}\right]\Gamma\left[h-\frac{i(k+\omega)}{2}\right]. \tag{137}$$

Substituting this inside equation (135) gives us the final expression for the Bogoliubov coefficient $\alpha_{\omega,k}$

$$\alpha_{\omega,k} = \frac{2^{2h} \cosh(\eta)^{-2h}}{N_{\omega,k}^* \sqrt{8\pi}\,\Gamma(2h)^2} \left[\frac{e^{\eta}-i}{e^{\eta}+i}\right]^{i\omega} \Gamma\left[h + \frac{i(k-\omega)}{2}\right]$$
$$\Gamma\left[h - \frac{i(k-\omega)}{2}\right]\Gamma\left[h + \frac{i(k+\omega)}{2}\right]\Gamma\left[h - \frac{i(k+\omega)}{2}\right]. \tag{138}$$

Repeating all these steps carefully, we can show that the Beta Bogoliubov coefficient is

$$\beta_{\omega,k} = -\frac{2^{2h} \cosh(\eta)^{-2h}}{N_{\omega,k} \sqrt{8\pi}\,\Gamma(2h)^2} \left[\frac{e^{\eta}+i}{e^{\eta}-i}\right]^{i\omega} \Gamma\left[h + \frac{i(k-\omega)}{2}\right]$$
$$\Gamma\left[h - \frac{i(k-\omega)}{2}\right]\Gamma\left[h + \frac{i(k+\omega)}{2}\right]\Gamma\left[h - \frac{i(k+\omega)}{2}\right]. \tag{139}$$

Referring to equation (112), we recall that the normalization constant appearing in the Bogoliubov coefficients is

$$N_{\omega,k} = \sqrt{\frac{\Gamma\left(h + \frac{i(\omega-k)}{2}\right)\Gamma\left(h + \frac{i(\omega+k)}{2}\right)\Gamma\left(h - \frac{i(\omega-k)}{2}\right)\Gamma\left(h - \frac{i(\omega+k)}{2}\right)}{2\omega\,\Gamma(2h)^2\Gamma(i\omega)\Gamma(-i\omega)}}. \tag{140}$$

With the explicit expressions in hand, one can check that the Bogoliubov coefficients satisfy the relations (59) and (60). This concludes our derivation of the Bogoliubov coefficients.

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
