# Peer review of "Bulk entanglement entropy in perturbative excited states"

_SciPost Physics, doi:SciPost Phys. 5, 024 (2018)_

## Round 2 · Referee Report · Anonymous · 2018-8-10

Strengths

1-Performing explicit computations. It is rare that the quantum corrections to the holographic entanglement entropy formula are explicitly computed.

Weaknesses

1-A minor weakness is that analytic computations are done only in the small interval limit.

2-The authors confirm only numerically that the change of entanglement entropy by an excitation in the CFT side agrees with the FLM formula.

Report

The authors compute the shift of entanglement entropy by an excitation in large c two-dimensional CFT, and compare the results to the bulk quantities proposed by Faulkner, Lewkowycz and Maldacena. In the AdS side, the bulk reduced density matrix is obtained in the mode expansion form, and the shift of the bulk entanglement entropy is computed in the small interval limit. The shift of the minimal area by back reaction is also evaluated. Combining them, the authors numerically find the agreement between the CFT result and the bulk result. It is a wonderful and important work. I recommend the paper for publication.

However, I would like to request the authors to fix some typos listed below.

Requested changes

1-In eq.(2.11), it seems that $\theta$ should be replaced by $\theta/n$. It is better to write the definition of $z$ in eq.(2.10) in terms of $\varphi$.

2-There should be $\frac{1}{1-n}\log$ in eq. (2.12). Also in (2.15), (2.17), (2.21) etc.

  • validity: high
  • significance: good
  • originality: high
  • clarity: high
  • formatting: excellent
  • grammar: excellent

Author:  Sagar F. Lokhande  on 2018-08-29  [id 310]

(in reply to Report 1 on 2018-08-10)
Category:
remark
correction

We thank the referee for their kind words and careful reading of the draft.

We agree with the assessment of weaknesses and hope that in future work we can overcome the technical difficulties in analytic continuations.

Regarding the requested changes: 1. Indeed, there is a 1/n missing in (2.11) which we have fixed. There was also a typo in (2.10) which we have corrected which we believe addresses the referee's concerns. 2. We have corrected these typos.

---

## Round 3 · List of Changes

1. Fixed a missing 1/n factor in equation (2.11). $theta$ now becomes $\theta/n$.
2. A typo in the definition of $z$ in equation (2.10) corrected and a few clarificatory words added just above this equation.
3. Forgotten factors of $\frac{1}{1-n}\log$ in equations (2.12), (2.15), (2.17) and (2.21) put in. These do not change any preceding or succeeding expressions.

You are currently on this page

Resubmission 1805.08782v3 on 5 September 2018

---

## Editorial Decision

published